# Nonlocal meta-lens with Huygens' bound states in the continuum

Jin Yao [1,8], Fangxing Lai[2,8], Yubin Fan[1,8], Yuhan Wang[2], Shih-Hsiu Huang[3], Borui Leng [1], Yao Liang [1], Rong Lin [1], Shufan Chen[1], Mu Ku Chen [1,4,5] ✉, Pin Chieh Wu [3,6,7] ✉, Shumin Xiao [2] ✉ & Din Ping Tsai [1,4,5] ✉

Meta-lenses composed of artificial meta-atoms have stimulated substantial interest due to their compact and flexible wavefront shaping capabilities, outperforming bulk optical devices. The operating bandwidth is a critical factor determining the meta-lens' performance across various wavelengths. Meta-lenses that operate in a narrowband manner relying on nonlocal effects can effectively reduce disturbance and crosstalk from non-resonant wavelengths, making them well-suitable for specialized applications such as nonlinear generation and augmented reality/virtual reality display. However, nonlocal meta-lenses require striking a balance between local phase manipulation and nonlocal resonance excitation, which involves trade-offs among factors like quality-factor, efficiency, manipulation dimensions, and footprint. In this work, we experimentally demonstrate the nonlocal meta-lens featuring Huygens' bound states in the continuum (BICs) and its near-infrared imaging application. All-dielectric integrated-resonant unit is particularly optimized to efficiently induce both the quasi-BIC and generalized Kerker effect, while ensuring the rotation-angle robustness for generating geometric phase. The experimental results show that the single-layer nonlocal Huygens' meta-lens possesses a high quality-factor of 104 and achieves a transmission polarization conversion efficiency of 55%, exceeding the theoretical limit of 25%. The wavelength-selective two-dimensional focusing and imaging are demonstrated as well. This work will pave the way for efficient nonlocal wavefront shaping and meta-devices.

Meta-lenses composed of subwavelength meta-atoms offer a means for compact and versatile manipulation of electromagnetic waves, with capabilities surpassing conventional bulk devices[1,2]. As a key factor for meta-lens, the operating bandwidth determines the meta-lens performance and potential application scenarios across various wavelengths[3,4]. For example, previous works have demonstrated the possibility of creating broadband metasurfaces with local electromagnetic responses for holographic imaging[5], bio-sensing[6], and polarization control[7]. Broadband achromatic meta-lenses that are effective for circularly polarized light and a variety of polarization

[1]Department of Electrical Engineering, City University of Hong Kong, Kowloon, Hong Kong SAR, China. [2]State Key Laboratory on Tunable Laser Technology, Ministry of Industry and Information Technology Key Lab of Micro-Nano Optoelectronic Information System, Shenzhen Graduate School, Harbin Institute of Technology, Shenzhen 518055, China. [3]Department of Photonics, National Cheng Kung University, Tainan 70101, Taiwan. [4]Centre for Biosystems, Neuroscience, and Nanotechnology, City University of Hong Kong, Kowloon, Hong Kong SAR, China. [5]State Key Laboratory of Terahertz and Millimeter Waves, City University of Hong Kong, Kowloon, Hong Kong SAR, China. [6]Center for Quantum Frontiers of Research & Technology (QFort), National Cheng Kung University, Tainan 70101, Taiwan. [7]Meta-nanoPhotonics Center, National Cheng Kung University, Tainan 70101, Taiwan. [8]These authors contributed equally: Jin Yao, Fangxing Lai, Yubin Fan. ✉e-mail: mkchen@cityu.edu.hk; pcwu@gs.ncku.edu.tw; shumin.xiao@hit.edu.cn; dptsai@cityu.edu.hk

states have also been reported. These meta-lenses can even facilitate full-color imaging by continuously tailoring the phase dispersion of meta-atoms[8–12]. Furthermore, the arrangement of these meta-lenses in arrays opens up possibilities for applications in high dimensional quantum light sources[13], as well as various multimodal perceptions, including depth sensing[14], edge detection[15], etc.

In contrast to broadband metasurfaces, resonances with a high quality factor (Q-factor) exhibit distinctive characteristics at specific wavelengths and effectively minimize interference and crosstalk from wavelengths that are not in resonance[16,17]. While local effects in metasurfaces can produce both broad and narrow responses[18], the optical response across multiple meta-atoms resulting from nonlocal effects can support high-Q resonances[19–22]. In fact, with the inclusion of additional spectral control and stable spatial phase modulation, there is significant interest in narrowband-operated wavefront shaping driven by nonlocal responses. Such advancements find applications in various fields, including optical nonlinearity, biomedical imaging, thermal radiation, and augmented reality/virtual reality display[23–28]. While many studies have successfully demonstrated high Q-factor responses associated with high field enhancement using metasurfaces, leveraging phenomena such as Fano resonance[29], non-radiating anapole mode[30], micro-cavity-integrated nanophotonics[31], and coherent lattice resonance[32], none of these methods have been able to simultaneously achieve both high-Q response and phase shift modulation. As an illustration, consider the phase gradient metasurface with subtle structural perturbations. This innovative technology has been proven to produce guided-mode resonances with an impressive Q-factor exceeding 2500 and the capability to steer light to desired precisely along a one-dimensional direction[33,34]. Another noteworthy advantage of high-Q metasurfaces is their capacity to achieve active tunability, even when the active material integrated into the system exhibits a relatively small index change in response to external stimuli[35]. As an instance, the characteristic of high-Q metasurface can be tuned through the electro-optic properties of lithium niobate[36] or the Kerr effect of silicon[37]. Recent advancements have also placed significant emphasis on quasi-bound states in the continuum (q-BICs) to achieve a high-Q response. These q-BICs have been incorporated into nonlocal metasurfaces for two-dimensional wavefront manipulation, which enables precise control over both the spatial and spectral characteristics of light and ultimately results in the experimental creation of a meta-lens boasting a high Q-factor of ~86 and a relatively low transmission conversion efficiency of around 4%[38,39]. Despite their limited operating bandwidth, the integration and stack of these nonlocal metasurfaces facilitate the extension of the spectral range, allowing for

independent control of multispectral responses. Their double-layer design with Fano resonances can further acquire higher efficiency and chiral response[40,41]. Nevertheless, nonlocal meta-lenses encounter the formidable challenge of finding the right balance between local phase control arising from the independent response of each meta-atom and nonlocal resonance excitation originating from the interactions between neighboring meta-atoms. As far as our current knowledge extends, there are no currently established strategies that can effectively address all of these aspects - Q-factor, efficiency, manipulation dimension, and footprint - simultaneously in the context of nonlocal wavefront shaping.

In this work, we experimentally demonstrate a nonlocal high-Q Huygens' meta-lens and explore its application in wavelength-selective imaging. Integrated-resonant unit (IRU)[42] is meticulously designed to effectively achieve a balance between Q-factor, efficiency, and the robustness of meta-atom's orientation, a challenge that was previously encountered. By introducing structural asymmetry in the configuration, we can excite the leaky q-BIC, resulting in a significantly higher Q-factor. The introduction of the generalized Kerker effect, which arises from the interference between dominant in-plane and out-of-plane magnetic dipole (MD) and other multipoles, plays a pivotal role in achieving high transmission efficiency. The robustness of IRU with respect to the rotation angle, which is responsible for generating geometric phase, is carefully engineered through geometry optimization to precisely control the near-field coupling between adjacent IRUs. By harnessing these effects, we strategically arrange the IRUs with geometric phases inside the nonlocal Huygens' meta-lens. This arrangement yields a relatively high Q-factor of 104 and an exceptional transmission conversion efficiency of 55%, exceeding the theoretical limit of 25%. Our experiments in focusing and imaging at different wavelengths strongly support the potential of our design for wavelength-selective optical applications.

## Results

### Design of Huygens' bound state in the continuum

Figure 1 schematically illustrates the distinction between meta-lenses with both efficient Q-factor and efficiency, transitioning from a local response to a nonlocal Huygens' response. When illuminated with broadband circularly polarized light, local dielectric meta-lenses can focus the entire spectrum without wavelength selectivity. This capability is accompanied by a high transmission polarization conversion efficiency $T_{LR}$ attributed to the high aspect ratio of its meta-atoms, which breaks the out-of-plane symmetry[10]. Here, $T_{LR}$ is defined as the ratio between the powers of polarization-converted light (right-

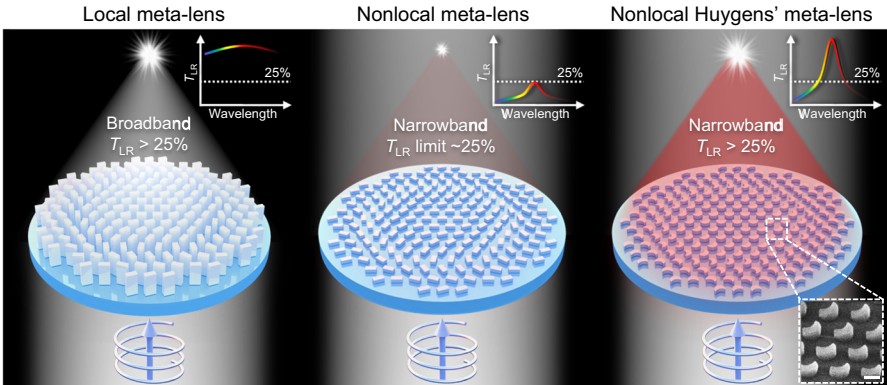

**Fig. 1 | Schematic illustration of local, nonlocal, and nonlocal Huygens' meta-lenses.** Local and nonlocal meta-lenses are generally limited by broadband responses and circular polarization conversion efficiency $T_{LR}$ of ~25%, respectively. Nonlocal Huygens' meta-lenses can simultaneously acquire narrowband wavefront shaping and efficiency $T_{LR}$ exceeding 25%. Inset: tilted scanning electron microscope (SEM) image of the fabricated sample. The scale bar is 500 nm. For simplicity, the local meta-lens discussed in this work comprises dielectric meta-atoms with heights at the scale of wavelengths, while the nonlocal meta-lenses are optically thin.

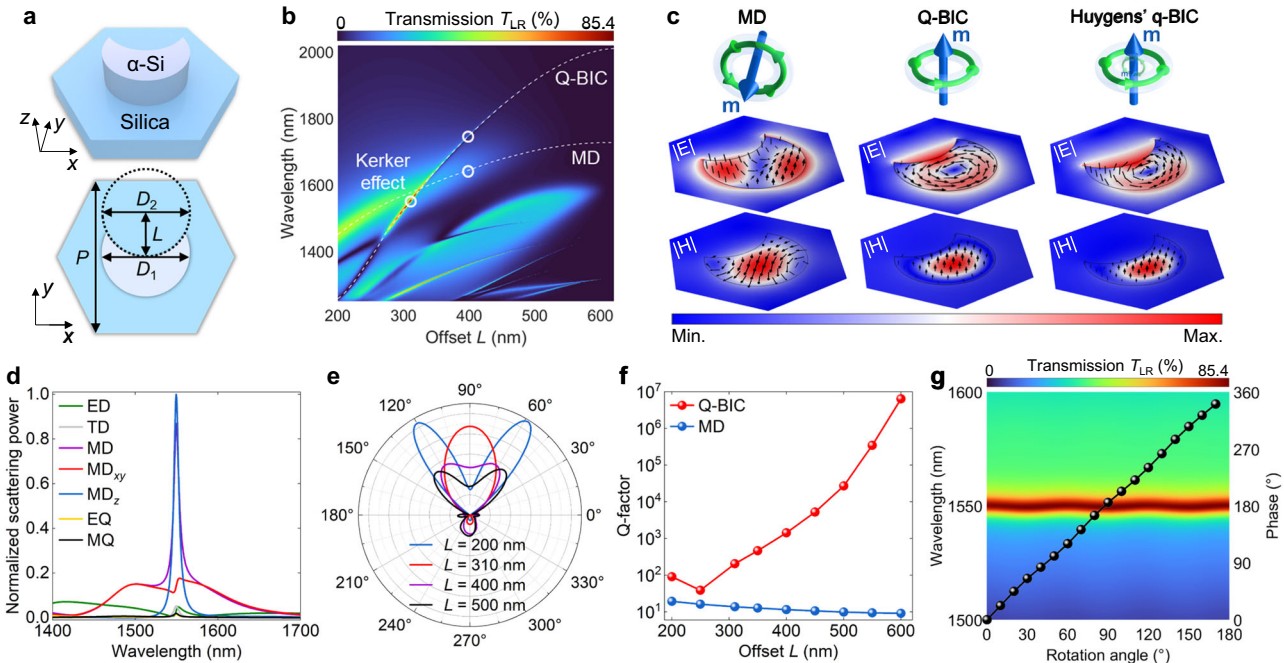

**Fig. 2 | Design of the IRU in nonlocal Huygens' meta-lens. a** Schematic illustration of the IRU. **b** Circular polarization transmission $T_{LR}$ as a function of offset $L$. White dashed lines indicate the resonant wavelengths of q-BIC and MD resonance. The white circle of the Kerker effect is extracted from the wavelength with the highest efficiency for $L = 310$ nm. **c** Simulated electromagnetic field distributions of two resonant modes and their hybridization. Black arrows denote the orientations of fields. The top panel schematically illustrates the corresponding mode. **d** Multipole decomposition of an IRU with $L = 310$ nm. **e** Far-field radiation patterns in the $xz$ plane for IRUs with $L = 200, 310, 400$, and $500$ nm. **f** Q-factors as a function of offset $L$. **g** Dependences of transmission $T_{LR}$ (color map) on the rotation angle of IRU. The black spheres represent the phase shift as a function of rotation angle at a wavelength of 1550 nm.

circularly polarized, RCP) to the incidence (left-circularly polarized, LCP). By harnessing the nonlocal effect arising from the interactions between meta-atoms, we achieve effective focusing solely on narrowband light, leaving light of non-resonant wavelengths unmodulated. Since introducing the nonlocal effect in meta-atoms with a high aspect ratio presents challenges, nonlocal meta-lenses are typically designed to be thin, lacking out-of-plane symmetry. Consequently, they function as a four-port system, with the transmissive circularly polarized conversion efficiency capped at ~25%[39,43–45] (refer to the middle image in Fig. 1). Further elaboration on the theoretical limit is provided in Supplementary Note 7. The Huygens' meta-lens, incorporating the generalized Kerker effect[46], emerges as a promising solution to surmount this limitation, particularly when the local response typically prevails. Consequently, it becomes imperative to design the nonlocal Huygens' meta-lens to deliver both a high Q-factor and efficiency, which would enhance the quality of focusing and imaging significantly.

Figure 2a gives a schematic of the building block of the proposed nonlocal Huygens' meta-lens. Each IRU consists of a crescent amorphous silicon nanopillar with a height of 327 nm placed on the silica substrate. The crescent shape is formed by trimming a circle with a diameter of 620 nm using another circle with the same size in diameter ($D_1 = D_2 = 620$ nm). The hexagonal lattice has a period of $P = 1000$ nm. The optical responses shown in Fig. 2 are solely based on simulations of periodic IRUs. With the normal incidence of LCP light from the substrate onto the IRU, two resonances can be excited: MD resonance and q-BIC. The former corresponds to a local Mie-type resonance with a typically low Q-factor, whereas the latter, benefitting from the structural asymmetry and the nonlocal effect, results in a high Q-factor. More details and discussions about the nonlocal effect are given in Supplementary Fig. 6. Figure 2b shows the transmission spectra of the designed IRUs. As the offset $L$ is varied, two resonances approach and interact with each other, resulting in the satisfaction of the generalized Kerker condition and thus achieving a high circular polarization conversion efficiency of $T_{LR} = 85.4\%$. This phenomenon is

primarily governed by the geometric parameter known as offset $L$. Furthermore, the introduction of offset disrupts structural symmetry, thereby introducing another level of freedom to generate orthogonal dipole moments and high-order multipoles. This complex coupling condition facilitates the achievement of polarization conversion functionality. Detailed transmission spectra are given in Supplementary Fig. 5. To better elucidate the achievement of polarization efficiency surpassing 25%, it is crucial to note that it is attributed to the synergistic engagement of both the Mie-type MD resonance and the q-BIC mode, rather than the disruption of out-of-plane symmetry. We explore this by varying the structural parameter of $L$ to modulate the resonant wavelengths of MD resonance and the q-BIC mode. As shown in Supplementary Fig. 7, we observe that the conversion efficiency remains capped at ~25% when the resonances of MD and q-BIC are spectrally distant, indicating the inherent weakness of out-of-plane asymmetry. More discussions about the out-of-plane symmetry can be found in Supplementary Note 8. According to the field distributions shown in Fig. 2c, we observe the presence of two types of circular electric fields and unidirectional magnetic fields at the wavelengths associated with the MD resonances. This observation suggests the excitation of in-plane and out-of-plane MD, respectively. At the wavelength where both of these resonances coexist (referred to as Huygens' q-BIC), the field distributions inherit characteristics from both resonances. This inheritance of characteristics presents the potential for generating interference between dominant MD and other multipole modes radiating to the forward direction, ultimately leading to the emergence of the generalized Kerker effect[46].

To provide further details about the characteristics of Huygens' q-BIC, Fig. 2d gives the multipole decomposition[47] of the IRU. At the resonant wavelength of ~1550 nm, we can observe low-Q in-plane and high-Q out-of-plane MD modes. These modes contribute to the dominant MD response, which is accompanied by a relatively weak electric dipole mode. Although out-of-plane MD cannot generate vertical radiation, other multipole moments induced by the q-BIC

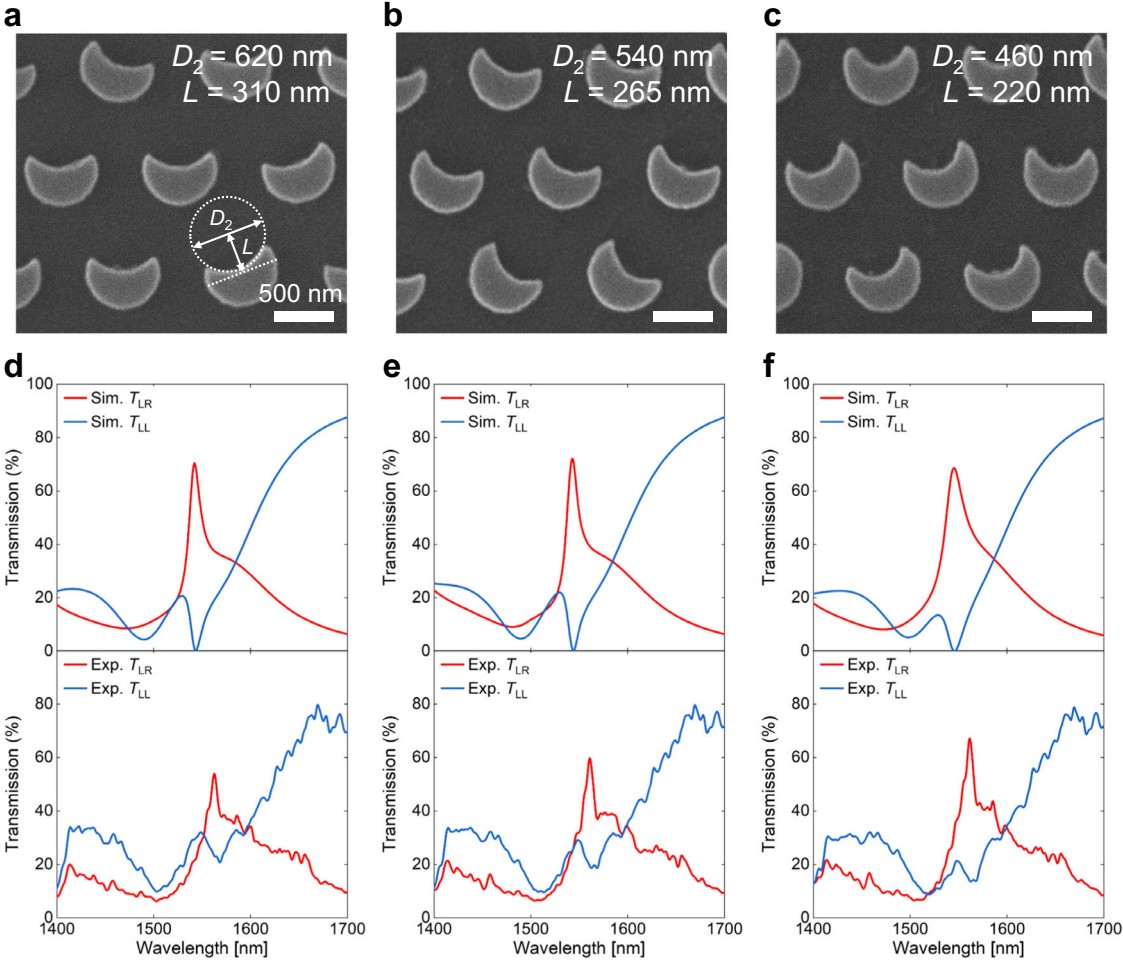

**Fig. 3 | Spectral responses of nonlocal Huygens' meta-lenses. a–c** SEM images of three fabricated nonlocal Huygens' meta-lenses. Scale bars are 500 nm. **d–f** Simulated and experimental transmission spectra $T_{LR}$ and $T_{LL}$ for the fabricated samples.

mode can influence the normal transmission, such as electric dipole, toroidal dipole, etc., as evidenced by peaks in the scattering spectra in Fig. 2d. The scattering spectra of horizontal multipoles are also given in Supplementary Fig. 11. These multipoles interfere with the in-plane MD induced by the low-Q MD resonance, creating the generalized Kerker effect that disrupts the radiation symmetry and enhances the transmission efficiency. Figure 2e shows the radiation pattern of the IRU with various offsets $L$, demonstrating the evolutionary process of the Kerker effect. The power has been normalized by the averaged power over all directions. Benefiting from the generalized Kerker effect, the sidelobes are diminished, and the radiation is directed predominantly along the $+z$ direction when $L = 310$ nm. The directivities[48] along the $+z$ direction for $L = 200$, 310, 400, and 500 nm are 1.26, 4.37, 2.34, and 1.41, respectively. This phenomenon serves to enhance transmissive efficiency. In terms of another crucial performance metric, the Q-factor can be flexibly adjusted by tuning the offset $L$, as shown in Fig. 2f. With the increase of the offset, the structural asymmetry is reduced, leading to a higher Q-factor for the q-BIC[49]. The abrupt shift observed at $L = 200$ nm is attributed to the influence of Wood's anomaly mode[50], which occurs at a wavelength of ~1256 nm. However, as this phenomenon does not impact our target functionality, it is omitted from further discussion for simplicity. The MD resonance remains relatively unaffected by changes in the offset, with only a slight decrease in the Q-factor observed. Given that the arrangement of the IRU in the meta-lens is based on the geometric phase[51], it becomes crucial to assess the robustness of the excited Huygens' q-BIC. Figure 2g presents the dependences of transmission

$T_{LR}$ and phase on the rotation angle of IRU. It is noteworthy that the resonant wavelength, Q-factor, and efficiency remain stable, with the phase being nearly twice the rotation angle. This observation suggests that the nonlocal Huygens' meta-lens is expected to maintain performance characteristics similar to those of the designed IRU.

## Characterizations of nonlocal Huygens' meta-lens

The designed IRUs are then arranged within the nonlocal Huygens' meta-lens in accordance with the spherical phase profile $\varphi$ as follows[52]:

$$\varphi = -\frac{2\pi}{\lambda}\left(\sqrt{f^2 + r^2} - f\right), \quad (1)$$

where $r$ is the radial coordinate, $\lambda = 1550$ nm is the working wavelength, and $f = 220$ μm is the focal length, thereby indicating a numerical aperture (NA) of 0.2. Figure 3a–c shows the SEM images of three fabricated samples. In these images, diameter $D_2$ takes on the values of 620, 540, and 460 nm, while the offset $L$ is set to 310, 265, and 220 nm, respectively. These parameter variations are implemented to ensure that these three meta-lenses possess different Q-factors while still maintaining the excitation of the Kerker effect. The simulated spectra of the corresponding periodic IRUs are presented in Supplementary Fig. 4. Figure 3d–f gives the simulated and experimental transmission spectra ($T_{LR}$ and $T_{LL}$) of these three meta-lenses. Notably, the experimental results align closely with the simulated results, demonstrating a strong agreement between the two datasets. Near the wavelength of 1550 nm, one can identify both the MD resonance and

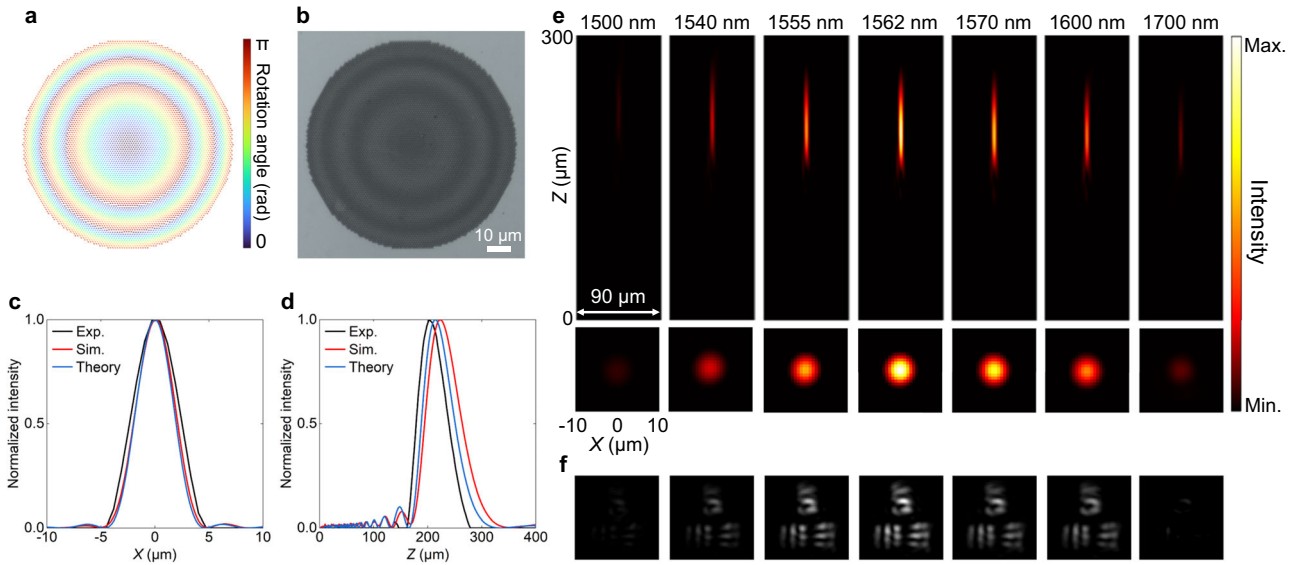

**Fig. 4 | Wavelength-selective focusing and imaging of nonlocal Huygens' meta-lens. a** Layout of meta-lens units with encoded rotation angle. **b** Optical micro-scopic image of the fabricated meta-lens as presented in Fig. 3a. The scale bar is 10 μm. **c, d** Experimental, simulated, and theoretical intensity distributions in the *x*-direction on the focal plane and *z*-direction at the resonant wavelength. **e,f** Experimental intensity distributions in *xz* and *xy* planes (**e**) and imaging (**f**) at different wavelengths.

Huygens' q-BIC by observing two resonance peaks in transmission spectrum $T_{LR}$, which correspond to modes with low and high Q-factors, respectively. The experimental (simulated) Q-factors of three nonlocal Huygens' meta-lenses are 104 (120), 98 (110), and 82 (71) with transmission conversion efficiencies of 55% (70%), 60% (72%), 69% (69%), respectively. We would like to point out that the Q-factor of 104 achieved in the nonlocal Huygens' meta-lens is higher than those reported in previous works on nonlocal meta-lens with q-BIC[39]. Furthermore, the efficiency has witnessed a large increase, surpassing the previous results by more than one order of magnitude. More comparisons with other similar works are given in Supplementary Table 1. In terms of Q-factor, efficiency, manipulation dimension, footprint, and fabrication requirements, our work stands out with superior performance when compared to previous studies (refer to Supplementary Note 14 for more details and discussions). The decrease in performance observed in the experiment can be attributed to nonradiative losses resulting from the imperfection, disorder, and finite size of the sample, which can be potentially improved by increasing the sample footprint and optimizing the fabrication technique[53]. Indeed, both the collimation and coherence of incident light can significantly impact optical performance. Supplementary Fig. 13 offers further discussions on the collimation and coherence of incident light, specifically in relation to the optical performance of the meta-lens.

The fabricated meta-lens as shown in Fig. 3a is then utilized to demonstrate its capability for wavelength-selective focusing and imaging. Figure 4a represents the layout of meta-lens units with encoded rotation angles, as per Eq. (1). Corresponding to Fig. 4a, Fig. 4b shows the optical microscopic image of the meta-lens, featuring a diameter of 90 μm. The intensity distributions in the *x*-direction on the focal plane at the resonant wavelength are given in Fig. 4c. Again, the experimental, simulated, and theoretical results exhibit a strong alignment with each other, indicating a close correspondence between the different sets of data. Their full-width half-maximums (FWHMs) are 4.8, 4.1, and 3.9 μm, respectively, implying a sub-diffraction-limited performance ($\lambda/2NA = 3.9$ μm) on the focal plane. Corresponding depths of focus along the *z* direction are 61.2, 70.3, and 62.8 μm, respectively, as shown in Fig. 4d. The deviation between these three results is due to imperfect amplitude, phase, and wavelength in practical operations

resulting from resonance excitations and fabrication imperfection. Figure 4e shows the experimental intensity distributions in the *xz* and *xy* planes at different wavelengths. Similar to the trend of the spectrum in Fig. 3d, the focusing intensity profile at the resonant wavelength shows maximum intensity, while those at non-resonant wavelengths are attenuated. This behavior is in good agreement with the simulation results (see Supplementary Fig. 14). The only noticeable effect is a wavelength deviation of approximately 20 nm, while the overall wavelength-selective property remains largely unaffected. The maximum focusing efficiency is around 47%. To demonstrate the practical application of imaging, the 1951 United States Air Force (USAF) resolution test chart was employed as the imaging target. The high Q-factor of the proposed meta-lens enables the demonstration of imaging with wavelength-selective characteristics, as shown in Fig. 4f. Leveraging the nonlocal effect, phase modulation exhibits superior performance and flexibility compared to the zone plate, which is constrained by a Q-factor of only around 20[54]. The reduced imaging quality results from the involvement of an oblique wavevector and inhomogeneous light spot for the structured light carrying image information, which cannot perfectly excite the resonance and produce the desired phase and amplitude. Additionally, the presence of coherent light speckles is inevitable. This concept of wavelength-selective wavefront shaping can be further extended to the visible band to realize the multispectral color router and augmented reality/virtual reality display (see details in Supplementary Fig. 15).

## Discussion

In summary, we have introduced the concept of a nonlocal Huygens' meta-lens that is composed of IRUs and demonstrated its utility in the realm of wavelength-selective imaging in the near-infrared region. By manipulating the geometric parameter of offset *L*, we can achieve the excitation and customization of q-BIC to attain a high Q-factor. Additionally, we can effectively generate the generalized Kerker effect through the interference between q-BIC, MD, and other multipole resonances, which significantly enhances the circular polarization conversion efficiency in transmission. The structure is also carefully designed to ensure the robustness of the IRU with respect to the rotation angle, which generates the geometric phase. Leveraging these effects, experimental results show

that the nonlocal Huygens' meta-lens can achieve both a high Q-factor of 104 and transmission conversion efficiency of 55%, exceeding the theoretical limit of 25%. The wavelength-selective imaging is demonstrated by using the 1951 USAF resolution test chart. This nonlocal Huygens' meta-lens can be utilized in active, integrated, and time-varying meta-devices[55–58], offering promising prospects for selective imaging in a real-time display system. The performance of our meta-lens can be further enhanced through the combination of both local and nonlocal phase controls, which can increase the degree of freedom for manipulating light in nanophotonics. The Q-factor in the proposed nonlocal Huygens' meta-lens might be further enhanced by incorporating high-order multipoles that exhibit lower radiation losses for meeting the conditions of the generalized Kerker effect[59]. We anticipate that the convergence of high-Q-factor, high-efficiency, two-dimensional, and compact wavefront shaping in nonlocal Huygens' meta-lens paves the way for narrowband-operated meta-lenses. This advancement facilitates their utilization in various applications such as nonlinear generation, quantum source, and augmented reality/virtual reality display.

## Methods

### Simulation
Electromagnetic responses of IRUs and meta-lenses were numerically simulated using commercial software COMSOL Multiphysics based on the finite element method. Perfectly matched layers (PMLs) were implemented at the top and bottom of the structure to truncate the open space. Periodic boundary conditions were applied in the $x$ and $y$ directions to simulate the periodic IRU. They were replaced by PMLs for the near-field simulation of meta-lenses. Far-field profiles were obtained by associating near-field simulation and scalar-diffraction-theory-based light propagation simulation. The refractive index of silicon is given in Supplementary Fig. 1, and that of silica substrate is 1.45.

### Fabrication
The fabrication process of the silicon meta-lens is given in Supplementary Fig. 2. First, the 327 nm α-Si film is deposited on the $SiO_2$ substrate (deposition rate 0.5 Å/s) and then covered by a 22 nm Cr film as a hard mask (deposition rate 0.5 Å/s) by an electron beam evaporator. Secondly, an 80 nm PMMA film is spin-coated and baked at 180 °C for an hour. Next, the PMMA resist was exposed to the electron beam (Raith E-line, 30 kV) and developed in MIBK/IPA solution for 30 s at 0 °C to form the PMMA nanostructures. After the resist development, the Cr and Si are successively etched by inductively coupled plasma (Oxford ICP180). Finally, the remaining Cr film is removed by immersing the sample into the chromium etchant for 10 min.

### Optical characterization
Optical measurement setups are plotted in Supplementary Fig. 3. Supercontinuum laser is used to provide wideband coherent light. Acousto-optic tunable filter is added to produce the single-wavelength light when measuring the light-field focusing and imaging. The circular polarization is generated by using a linear polarizer and a quarter-wave plate. A lens and an object (20× magnification, NA = 0.4) are employed to generate the collimated incident light. An object (20× magnification, NA = 0.4) is used to collect the transmission light. After passing through the quarter-wave plate and the linear polarizer, corresponding left-handed and right-handed circular components can be analyzed by spectrometer or camera.

## Data availability
Data underlying the results are available from the corresponding authors upon request.

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

## Acknowledgements

This work is supported by the University Grants Committee / Research Grants Council of the Hong Kong Special Administrative Region, China [Project No. AoE/P-502/20, CRF Project: C1015-21E; C5031-22G, GRF Project: CityU15303521; CityU11305223; CityU11310522; CityU11300123, and Germany/Hong Kong Joint Research Scheme: G-CityU 101/22], City University of Hong Kong [Project No. 9380131, 9610628, and 7005867], and National Natural Science Foundation of China [62375232]. S.X. acknowledges financial support from National Key R&D Program of China (Grant Nos. 2021YFA1400802), the National Natural Science Foundation of China (Grant Nos. 62125501, and 6233000076), Fundamental Research Funds for the Central Universities (Grant No. 2022FRRK030004), and Shenzhen Fundamental Research Projects (Grant Nos. JCYJ20220818102218040). P.C.W. acknowledges the support from the National Science and Technology Council (NSTC), Taiwan (Grant number: 111-2112-M-006-022-MY3; 111-2124-M-006-003; 112-2124-M-006-001), and in part from the Higher Education Sprout Project of the Ministry of Education (MOE) to the Headquarters of University Advancement at NCKU. P.C.W. also acknowledges the Yushan Fellow Program by the MOE, Taiwan for the financial support. The research is also supported in part by Higher Education Sprout Project, Center for Quantum Frontiers of Research & Technology (QFort) at NCKU. We thank Prof. Hau Ping Chan for optical device support.

## Author contributions

J.Y. and D.P.T. conceived the idea. J.Y. designed the samples and carried out the theoretical simulations. F.L., Y.W., and S.X. conducted the sample fabrications. J.Y., Y.F., S.H.H., B.L., and S.C. performed the optical measurements. J.Y., Y.L., R.L., M.K.C., and P.C.W. performed the data analysis. J.Y. prepared the manuscript, and all authors reviewed it. M.K.C., P.C.W., S.X., and D.P.T. initiated and supervised the research.

## Competing interests

The authors declare no competing interests.
