## [Peer Review File · Nature Communications]

Nonlocal meta-lens with Huygens' bound states in the continuumREVIEWER COMMENTS

Reviewer #1 (Remarks to the Author):

The article demonstrates a metasurface in transmission mode for focusing with the idea to enable a broadband performance. The authors show results experimentally. While the work is interesting there are some issues that the reviewer cannot suggest publication:

- The work is not very novel. The building block and polarization control is a known topic and already several people have used for various applications
- By broadband the authors show results for only 1540-1600 nm (which is not multi-wavelength spectrum)
- The transmission efficiency is only 50%

Given these, while the work is interesting, thy reviewer cannot suggest publication in this Journal.

Reviewer #2 (Remarks to the Author):

This paper presents and experimentally validates a novel non-local metalens that simultaneously achieves high Q and high efficiency in the IR range. The authors also show how the lens can be used for selective wavelength imaging. The concept and the design of the metalens are intriguing and valuable for researchers in this field. I recommend that this manuscript be accepted for publication in Nature Communications after addressing the following points:

1- The manuscript does not explain how the design process was conducted or why the chosen shape for the unit cell is better than other alternatives. For instance, what are the reasons for selecting a hexagonal lattice and a crescent shape for the unit cells? What benefits do they offer? The supplementary material compares different shapes for the unit cell, but it is not well referenced in the manuscript.

2- The manuscript does not provide a clear and comprehensive comparison with other similar works. For example, what is the advantage of this work over [Lin L., et al, Universal narrowband wavefront shaping with high quality factor meta-reflect-arrays. Nano Lett. 23, 1355-1362 (2023)] ? The design in that paper provides higher Q-factor and comparable efficiency when compared with this work.

3- The paper claims that the lens provides a sub-diffraction-limited full-width half-maximums (FWHMs). But it seems that this claim is only based on the results shown in the x-y plane (Fig. 4b). The data from the x-z plane (Fig. 4c) suggests that the FWHM is much larger in that direction. To clarify this point, I recommend that the authors plot the data from Fig. 4c in a 1D graph as well, as they did for the x-y plane.

4- Comparing the experimental results reported in Fig. 4 of the manuscript to the simulation results reported in Fig. 10 of the supplementary, there is a considerable shift in the operating wavelength. Considering the fact that the lens is designed to be very high-Q and therefore narrowband, this shift is important and need to be addressed and explained in the manuscript.

Reviewer #3 (Remarks to the Author)

The authors have done a solid job. However, some results and their interpretation raise questions. I do not recommend the manuscript for publication in the current form.

I encourage the authors to address the following questions:

1. Could the Authors, briefly explained the difference local and non-local metasurfaces and also explain in simple words why the theoretical limit of polarization conversion is 25% for nonlocal metasurfaces and there is no such a limit for local metasurfaces? Please also defined the efficiency of polarization conversion in the text. The cited paper Ref. 35 does not explain the limit of 25% but it addresses authors the readers to other papers of the same authors.

3. As I know, the overcoming the limit of the polarization conversion of 25% requires the breaking out-of-plane symmetry. Due to the substrate, the out-of-plane symmetry is already broken and, thus, this limit is not applicable to the structures in Fig. 2. Therefore, strictly speaking, Figure 2 is not correct.

4. The inset in Fig. 1 illustrating the fabricated sample is unclear. It is quite difficult to understand the difference between the initial design and the fabricated sample.

5. The design of metasurface is unclear from the provided figures. If I understood correctly, the structure is periodic, and all the meta-atoms are identical. The only difference between the unit cells of the metalens is the rotation angle of the crescent shape meta-atoms. It would be illustrative if the authors encode the orientation of the crescent shape meta-atoms with colors and plot the design of the metalens (top view)

6. Why do the Authors need q-BIC and high Q-factor? How does the high-Q factor affect the performance of metalens? High-Q factor shows the near-field enhancement but does not affect the far-field field enhancement. Why do the Authors need near-field enhancement due to quasi-BIC and high-Q factor?

7. Vertical and horizontal magnetic dipoles cannot result in the enhanced scattering to forward direction as the vertical magnetic dipole does not radiate to the vertical direction. Therefore, it cannot affect the transmission at normal incidence.

8. Figure 2b shows the transmission map. The maximum transmission is about 85%, but this value is almost indistinguishable in the figure.

9. Am I right that Figure 2 is plotted for periodic structure but not for the metalens? This is unclear from the text of the paper.

10. What is the physical mechanism responsible for the polarization conversion? How was circular polarization transmission $T_{\{LR\}}$ optimized to achieve the maximal value? Blind numerical optimization?

11. Usually, the Q-factor of q-BIC decreases with the increase in the asymmetry parameter but Figure 2f shows the inverse behavior. What is the reason for such a behavior? Is this resonance indeed a q-BIC? What are the benefits from a metalens with high-Q factor?

12. Can the authors plot directivity in Fig. 2D, i.e. intensity along each direction normalized by the intensity averaged over all directions?

13. The nonlocal effect, which leads to the generation of Q-BIC, plays a crucial role in the proposed nonlocal Huygens' metalens. However, this effect is highly sensitive to the surrounding nanostructures. To illustrate the nonlocal effect in the proposed nonlocal Huygens' metalens, the authors analyze the multipole modes supported in the periodic nanostructure with varying radial unit numbers. Nevertheless, if these nanostructures are employed to create a metalens with a non-periodic array due to the geometric phase, how do the nanostructures with different orientations affect the generation of Q-BIC and the working wavelength?

14. Is it possible to expand the applicability of the proposed nonlocal Huygens' metalens to work under ambient illumination? This is crucial considering the potential applications of the color router as shown in Supplementary Fig. 11: which typically operates in natural environments under ambient conditions.

For Reviewer 1:

General Comment:

The article demonstrates a metasurface in transmission mode for focusing with the idea to enable a broadband performance. The authors show results experimentally. While the work is interesting there are some issues that the reviewer cannot suggest publication:

Given these, while the work is interesting, thy reviewer cannot suggest publication in this Journal.

Response:

We thank the Reviewer for his/her comments. In our work, we propose the nonlocal Huygens' meta-lens to achieve the two-dimensional wavefront shaping with simultaneously high Q-factor (i.e. **narrowband** instead of broadband) and high efficiency. The novelty, challenge, design, performance parameters, and potential applications of nonlocal (narrowband) meta-lenses are intrinsically distinct from broadband meta-lenses with local responses. To make it clearer, more explanations and discussions are given in Responses to Specific Comments below.

Specific Comments:

1. The work is not very novel. The building block and polarization control is a known topic and already several people have used for various applications.

Response:

We thank the Reviewer's comment on this point. To begin with, it is important to highlight the value of exploring different methods and approaches that achieve similar optical functionalities. For instance, when it comes to polarization control using metasurfaces, there exists a variety of strategies. These include a combination of geometric phase control and propagation phase [*Nat. Nanotechnol.* 13, 227-232 (2018)], utilization of anisotropic nanostructures [*Nano Lett.* 20, 6351-6356 (2020)], and harnessing coupling between active materials and chiral meta-atoms [*Nat. Mater.* 20, 1024-1028 (2021)], all of which hold significance in their own right and find application in distinct scenarios.

As discussed in the first paragraph of Page 3 in the **Main Article**, broadband meta-lenses with local responses have been extensively reported due to their wide range of applications. Designing and implementing their building block and controlling polarization are relatively straightforward tasks, achieved by manipulating the independent response of individual meta-atoms. In contrast, narrowband meta-lenses with nonlocal responses serve specific applications. However, designing meta-atoms and controlling polarization in such cases pose challenges due to the need to balance local phase control, driven by

individual meta-atom responses, with nonlocal resonance excitation resulting from interactions between neighboring meta-atoms. Currently, there are no established strategies capable of effectively addressing all these aspects - including Q-factor, efficiency, manipulation dimension, and footprint - simultaneously in the context of nonlocal wavefront shaping. Our work has put forward an efficient approach and has experimentally demonstrated a meticulously designed metasurface to overcome these challenges. The novelty and advances of our work are listed and elaborated as follows.

(1) Novel design of integrated-resonant units. The design of integrated-resonant units (IRUs) should consider three key points:

- (a) High Q-factor. By breaking the in-plane symmetry of the structure, we introduce symmetry-protected quasi-bound states in the continuum (q-BICs) to boost the Q-factor. This enhancement can be flexibly manipulated by tailoring the asymmetric parameter.
- (b) Robustness of the rotation angle for generating geometric phase. Rotation angle is used to generate the geometric phase for the metasurface. Resonance characteristics such as Q-factor, efficiency, phase, and resonant wavelength must remain robust across the rotation angles to ensure both effective phase modulation and resonance excitation simultaneously.
- (c) High transmission efficiency. In addition to the q-BIC mode, we introduce a Mie-type magnetic dipole (MD) resonance to couple with q-BIC and induce the generalized Kerker effect. This effect facilitates unidirectional transmission and enhances the efficiency of transmissive circular polarization conversion, surpassing the theoretical limit of 25%.

Simultaneously achieving the three key points mentioned above is challenging to realize in conventional meta-units with a single resonant mode and mediocre structure. Additionally, we would like to highlight that two types of balances need to be considered during the optimization of IRU (also refer to Supplementary Fig. 8):

- i. The optimization of geometric shape involves striking a balance between symmetry, which ensures rotation robustness, and asymmetry, which facilitates the excitation of leaky q-BIC excitation.
- ii. The optimization of array periods requires balancing weak coupling (achieved with a larger period) for rotation robustness against strong coupling (achieved with a smaller period) between IRUs for realizing a high Q-factor.

The successful demonstration of a meta-unit design that fulfills these key points underscores the value and novelty of our approach.

(2) Strong performance parameters. We simultaneously consider four

aspects - Q-factor, efficiency, manipulation dimension, and footprint - to demonstrate the high performance of our work. Recent research on single-layer nonlocal meta-lens for two-dimensional wavefront shaping based on q-BICs achieves a Q-factor of ~ 86 and a transmission conversion efficiency of $\sim 4\%$ [*Light: Sci. Appl.* 11, 246 (2022)]. In contrast, our proposed nonlocal Huygens' meta-lens achieves a Q-factor of 104 and a significantly improved transmission conversion efficiency of 55%. This represents a remarkable increase in efficiency by one order of magnitude. In addition, compared with [*Nat. Nanotechnol.* 15, 956-961 (2020)] and [*Nano Lett.* 23, 1355-1362 (2023)], our work possesses the capability of two-dimensional wavefront shaping, which is a challenging feat due to the considerable influence of the nonlocal effect on the wavefront shaping dimension. Compared with some simulation works ([*Phys. Rev. Lett.* 126, 073001], [*Adv. Photonics* 3, 026002], [*Nano Lett.* 20, 5127-5132 (2020)], and [*Nano Lett.* 22, 1703-1709 (2022)]), our work does not need precise alignment of two layers or a robust fabrication process with precision up to a few nanometers. More detailed comparisons are provided in Table R1 (i.e. newly added Supplementary Table 1). When considering the four aspects of Q-factor, efficiency, manipulation dimension, and footprint, our work demonstrates superior performance compared with previous works.

Table R1. Performance comparison between meta-devices for nonlocal wavefront shaping.

References	Quality factor	Efficiency	Wavefront shaping dimension	Number of layer
This work (Exp.)	104	55%	2D	1
This work (Sim.)	120	70%	2D	1
Light: Sci. Appl. 11, 246 (2022) (Exp.)	~ 86	$\sim 4\%$	2D	1
Nat. Nanotechnol. 15, 956-961 (2020) (Exp.)	>2500	$>20\%$	1D	1
Nano Lett. 23, 1355-1362 (2023) (Exp.)	~ 380	59%	1D	1
Phys. Rev. Lett. 125, 017402 (Sim.)	~ 120	$\sim 25\%$	2D	1
Phys. Rev. Lett. 126, 073001 (Sim.)	~ 500	$>90\%$	2D	2
Adv. Photonics 3, 026002 (Sim.)	~ 500	$>90\%$	2D	2
Nano Lett. 20, 5127-5132 (2020) (Sim.)	~ 3600	$\sim 60\%$	1D	1
Nano Lett. 22, 1703-1709 (2022) (Sim.)	~ 30000	$>80\%$	1D	1

(3) Wavelength-selective two-dimensional focusing and imaging. We demonstrate wavelength-selective two-dimensional focusing, achieving sub-diffraction-limited performance. To advance practical application, we experimentally implement wavelength-selective imaging. Up to this point,

the realization of wavelength-selective two-dimensional focusing and imaging has only been reported in [*Nano Lett.* 23, 6768-6775 (2023)], albeit with a Q-factor of ~ 20 , relying on a mechanism based on the zone plate without phase modulation. Furthermore, we introduce the concept of potential applications in multispectral augmented reality/virtual reality displays, underscoring the versatility of our approach. Discussions and analysis are provided on Pages 11-14 in the **Main Article** and Page 21 in **Supplementary Note 15**.

To enhance the novelty and significance of the metasurface design in our approach, the following discussions have been added to the Supplementary in **Supplementary Note 9**:

"In the main article, we delve into the realm of meta-lenses, particularly focusing on the widespread exploration of broadband varieties with local responses. These lenses, prized for their versatility, often boast straightforward designs and polarization control, thanks to the independent manipulation of individual meta-atoms. However, our attention shifts to narrowband meta-lenses with nonlocal responses, tailored for specific applications. Herein lies the challenge: striking a delicate balance between harnessing individual meta-atom responses for local phase control and navigating the complexities of nonlocal resonance excitation arising from interactions among neighboring meta-atoms. Presently, no established strategies exist to comprehensively address the myriad facets - Q-factor, efficiency, manipulation dimension, and footprint - simultaneously within the context of nonlocal wavefront shaping. Our work endeavors to tackle this conundrum with a novel approach, exemplified by a meticulously crafted metasurface. Let us delve deeper into the pioneering aspects and advancements of our methodology on the pioneering design of IRU (also refer to Supplementary Fig. 8).

(a) Amplified Q-factor: By disrupting the structure's in-plane symmetry, we usher in symmetry-protected q-BICs, lending a significant boost to the Q-factor. This augmentation is achieved by fine-tuning the asymmetric parameter, affording flexibility in manipulation.

(b) Stalwart rotation angle robustness: The rotation angle assumes paramount importance in generating the geometric phase essential for meta-lens functionality. Ensuring the steadfastness of resonance characteristics - Q-factor, efficiency, phase, and resonant wavelength - across varying rotation angles is imperative for seamless phase modulation and resonance excitation.

(c) Enhanced transmission efficiency: Beyond the q-BIC mode, our innovation encompasses the incorporation of a Mie-type MD resonance. This synergistic coupling with q-BIC engenders the generalized Kerker effect,

fostering unidirectional transmission and surpassing the theoretical limit of 25% for transmissive circular polarization conversion.

Concurrently attaining these three pivotal points presents a formidable challenge within the realm of conventional meta-units characterized by a single resonant mode and lackluster structure. Moreover, the optimization of IRUs warrants a delicate equilibrium, wherein geometric shape optimization necessitates a harmonious interplay between symmetry and asymmetry to cater to rotation robustness and the excitation of leaky q-BIC excitation. Selecting a lattice and unit with improved symmetry while maintaining asymmetry, such as a hexagonal lattice, optimizes this balance. The crescent shape, derived from a circle, is chosen for its superior symmetry, and the optimization process is detailed in Supplementary Fig. 9. Similarly, the optimization of array periods demands a judicious equilibrium, balancing weak coupling for rotation robustness with strong coupling between IRUs to realize a high Q-factor (refer to Supplementary Fig. 10)."

Supplementary Fig. 8: Design flow for an IRU with high Q-factor, rotation robustness, and high efficiency.

Furthermore, we have enhanced the clarity and comprehensiveness of the relevant descriptions on Pages 11 and 12 of the **Main Article** to facilitate a clearer and more thorough comparison.

"More comparisons with other similar works are given in Supplementary Table 1. In terms of Q-factor, efficiency, manipulation dimension, footprint, and fabrication requirements, our work stands out with superior performance when compared to previous studies (refer to Supplementary Note 14 for more details and discussions)."

and on Page 14 of the **Main Article**:

“Leveraging the nonlocal effect, phase modulation exhibits superior performance and flexibility compared to the zone plate, which is constrained by a Q-factor of only around 20⁵⁴.”

In addition, we have incorporated a thorough comparison and discussion in **Supplementary Note 14** to underscore the advantages of our approach and highlight the distinctions from other works.

“We have compared our work with Ref. [5] in the main text because both two works are single-layer nonlocal meta-lenses for two-dimensional wavefront shaping. To provide a more comprehensive comparison, a comparison with another similar work of Ref. [7] is also given as follows. It proposed a novel phase gradient metasurface made entirely from individually addressable high-Q-factor meta-atoms. Although the Q-factor of ~380 reported in Ref. [7] is higher than our work and the efficiency of 59% is comparable, it is important to highlight the advantages and distinctions of our work as follows:

(a) Advantages

- i. Our work achieves two-dimensional wavefront shaping with high Q-factor and efficiency, surpassing the previous work’s demonstration of only one-dimensional control⁷. The addition of wavefront shaping dimension significantly impacts optical performance, while the limited manipulation dimension constrains versatile applications.
- ii. Unlike previous work, our fabrication process doesn't demand extreme precision. In Fig. 2b in the main article, the q-BIC mode and Kerker effect exhibit stability even with slight wavelength deviations in changing geometric parameters. In contrast, previous work requires precise control over geometric parameters for wavefront engineering⁵, posing challenges for practical implementation due to the need for a fabrication process with precision up to a few nanometers.
- iii. Transmissive optical components are more commonly used in practical applications. The involvement of a metallic back reflector in Ref. [7] confines the metasurface to operate only in a reflection scheme, limiting its range of applications. In contrast, our metasurface operates in a transmission scheme with 2D wavefront control capability, making it more promising for real-world applications.

(b) Differences

The physical mechanisms between our work and previous work are inherently distinct. In our approach, we leverage q-BIC and the Kerker effect to achieve both high Q-factor and efficiency, along with optimized rotation robustness for the geometric phase. In contrast, the reported work manipulates the geometric parameters of meta-atoms to control the resonance phase of guide-mode resonances. Additionally, the incident polarization states differ, with our work utilizing circular polarization and the reported work employing linear polarization. Despite these differences, both approaches offer a platform for efficient high-Q wavefront shaping.”

2. By broadband the authors show results for only 1540-1600 nm (which is not multi-wavelength spectrum).

Response: We thank the Reviewer for his/her comments. First, it is important to clarify that our primary objective does not entail realizing a meta-lens working across a multi-wavelength spectrum. As previously emphasized, achieving a narrowband meta-lens with nonlocal responses presents considerable challenges. Therefore, our main aim is to develop a **narrowband** meta-lens rather than a broadband one, with a focus on achieving high efficiency within a specific working band. Compared to previous works on nonlocal meta-lenses with q-BIC (see *Light: Sci. Appl.* 11, 246 (2022)), our study demonstrates a significantly higher Q-factor, along with a remarkable increase in efficiency by more an order of magnitude. The narrowband characteristic is clearly illustrated by the narrow transmission T_{LR} peak in Fig. 3d-f with the wavelength range of 1400-1700 nm. Figure 4 shows the wavelength-selective (i.e. narrowband) property of the proposed nonlocal Huygens' meta-lens. In this representation, the central wavelength exhibits the highest efficiency, while efficiencies at other wavelengths are dramatically reduced. Additionally, we demonstrate the multi-wavelength attribute by integrating three types of IRUs in the visible spectrum, as discussed on Page 21 in **Supplementary Note 15**. These underscore the importance and uniqueness of our work in addressing the challenges associated with narrowband meta-lenses with nonlocal responses.

3. The transmission efficiency is only 50%.

Response: We thank the Reviewer's comment on this point. Below, we outline three points that underscore the high efficiency of 55% in 2D nonlocal wavefront shaping:

- (a) Theoretical limit. As illustrated in Figure 1, nonlocal meta-lenses are typically designed to be optically thin for resonance excitations. They can be approximated as a four-port system, where the transmissive circular polarization conversion efficiency is constrained to a maximum of ~25%.
- (b) Challenge in combining local phase control and nonlocal resonance excitation. As discussed on Pages 3 and 4 in the **Main Article**, nonlocal meta-lenses pose challenges due to the delicate balance required between local phase control and nonlocal resonance excitation. While an efficiency of 55% may not be considered high in conventional broadband and local meta-lenses, it is intrinsically high for nonlocal meta-lenses, especially when considering other aspects such as high Q-factor, two-dimensional wavefront shaping, and single-layer construction.
- (c) Comparison with reported works. Previous studies have reported

transmission conversion efficiency of ~4% for single-layer nonlocal meta-lenses with q-BICs, achieving two-dimensional wavefront shaping (*Light: Sci. Appl.* 11, 246 (2022)). In contrast, our proposed nonlocal Huygens' meta-lens achieves a transmission conversion efficiency of 55%, representing a remarkable increase by an order of magnitude. Further comparison details are provided in Table R1 (i.e. Supplementary Table 1).

To improve clarity regarding the significance of the achieved transmission efficiency for 2D nonlocal wavefront control, we have revised the descriptions concerning the polarization conversion limit on Pages 5 and 6 of the **Main Article**:

“When illuminated with broadband circularly polarized light, local dielectric meta-lens can focus the entire spectrum without wavelength selectivity. This capability is accompanied by a high transmission polarization conversion efficiency T_{LR} attributed to the high aspect ratio of its meta-atoms, which breaks the out-of-plane symmetry¹⁰. Here, T_{LR} is defined as the ratio between the powers of polarization-converted light (right-circularly polarized, RCP) to the incidence (left-circularly polarized, LCP). By harnessing the nonlocal effect arising from the interactions between meta-atoms, we achieve effective focusing solely on narrowband light, leaving light of non-resonant wavelengths unmodulated. Since introducing the nonlocal effect in meta-atoms with a high aspect ratio presents challenges, nonlocal meta-lenses are typically designed to be thin, lacking out-of-plane symmetry. Consequently, they function as a four-port system, with the transmissive circularly polarized conversion efficiency capped at ~25%^{39,43-45} (refer to the middle image in Fig. 1). Further elaboration on the theoretical limit is provided in Supplementary Note 7.”

with additional references

“43. Monticone F., Estakhri N. M., Alu A. Full control of nanoscale optical transmission with a composite metascreen. *Phys. Rev. Lett.* **110**, 203903 (2013).”

44. Ding X., Monticone F., Zhang K., Zhang L., Gao D., Burokur S. N., De Lustrac A., Wu Q., Qiu C. W., Alù A. Ultrathin Pancharatnam-Berry metasurface with maximal cross-polarization efficiency. *Adv. Mater.* **27**, 1195-1200 (2015).

45. Hassanfiroozi A., Huang P. S., Huang S. H., Lin K. I., Lin Y. T., Chien C. F., Shi Y., Lee W. J., Wu P. C. A Toroidal-Fano-resonant metasurface with optimal cross-polarization efficiency and switchable nonlinearity in the near-infrared. *Adv. Opt. Mater.* **9**, 2101007 (2021).”

The caption of Fig. 1 is also revised to clarify the difference between local and nonlocal meta-lenses:

“For simplicity, the local meta-lens discussed in this work comprises dielectric meta-atoms with heights at the scale of wavelengths, while the nonlocal meta-lenses are optically thin.”

The following discussions regarding the theoretical limit of polarization conversion efficiency are added on Page 8 in **Supplementary Note 7**:

“As discussed in the main context, the metasurface can be conceptualized as a four-port system. In such cases, the transmission coefficient of cross-polarization conversion t_{LR} can be derived as^{1,2}:

$$|t_{LR}|^2 = \text{Re}[t_{LL}] - |t_{LL}|^2 \quad (\text{S1})$$

Here, t_{ij} represents the transmission coefficient for i -polarized light when the metasurface is illuminated with j -polarized light. The subscripts L and R signify left-handed polarization and right-handed polarization, respectively. Referring to Eq. (S1), the maximum cross-polarization conversion efficiency $T_{LR} = |t_{LR}|^2$ reaches 25% when $t_{LL} = 50\%$.”

For Reviewer 2:**General Comment:**

This paper presents and experimentally validates a novel non-local metalens that simultaneously achieves high Q and high efficiency in the IR range. The authors also show how the lens can be used for selective wavelength imaging. The concept and the design of the metalens are intriguing and valuable for researchers in this field. I recommend that this manuscript be accepted for publication in Nature Communications after addressing the following points:

Response:

We thank the Reviewer for his/her positive comments and constructive suggestions on our manuscript. We addressed the comments point-to-point as follows.

Specific Comments:

1. The manuscript does not explain how the design process was conducted or why the chosen shape for the unit cell is better than other alternatives. For instance, what are the reasons for selecting a hexagonal lattice and a crescent shape for the unit cells? What benefits do they offer? The supplementary material compares different shapes for the unit cell, but it is not well referenced in the manuscript.

Response:

We thank the Reviewer for the comment. To develop a meta-lens that possesses high Q-factor and high transmission efficiency using nonlocal effect, several conditions must be carefully considered. Firstly, the design should prioritize breaking the in-plane symmetry of the unit to engage symmetry-protected quasi-bound states in the continuum (q-BICs), a crucial step in maximizing the Q-factor. Secondly, robustness in the rotation angle for generating the required geometric phase is essential to maintain resonance characteristics such as Q-factor, efficiency, and wide phase shift across different angles. This ensures effective phase modulation and resonance excitation. Lastly, the introduction of additional resonances, such as Mie-type magnetic dipoles (MD), can further enhance efficiency by exploiting nonlocal effects like the generalized Kerker effect. These considerations collectively contribute to achieving optimal performance in integrated-resonant units (IRUs) for high Q-factor and transmission efficiency.

Consequently, as illustrated in Supplementary Fig. 8, the design process of integrated-resonant units (IRUs) entails three sequential considerations:

(a) High Q-factor. Disrupting in-plane symmetry enhances the Q-factor by engaging q-BICs, which can be adjusted by tailoring the asymmetric parameter. Various unit types with asymmetry, including both square and hexagonal lattices, can fulfill this requirement.

(b) Robustness of the rotation angle for generating geometric phase. Rotation is used to generate the geometric phase required by the meta-lens. Resonance characteristics must remain stable across different angles, necessitating a balance between geometric shape optimization for rotation robustness and array period optimization for coupling strength. Selecting a lattice and unit with improved symmetry while maintaining asymmetry, such as a hexagonal lattice, optimizes this balance. The crescent shape, derived from a circle, is chosen for its superior symmetry, and the optimization process is detailed in Supplementary Fig. 9. Subsequently, the period is optimized considering Q-factor, efficiency, and rotation robustness, as shown in Supplementary Fig. 10.

(c) High efficiency. Introducing another Mie-type magnetic dipole (MD) resonance enhances by coupling with q-BIC, generating the generalized Kerker effect. This effect enables unidirectional transmission and enhances circularly polarized conversion efficiency, surpassing theoretical limits. Performance optimization in this step primarily involves adjusting geometric parameters while keeping the shape fixed.

To address the reviewer's concern and enhance the novelty and significance of the metasurface design in our approach, we have added the following discussions to the main article and supplementary.

Supplementary Note 9:

"In the main article, we delve into the realm of meta-lenses, particularly focusing on the widespread exploration of broadband varieties with local responses. These lenses, prized for their versatility, often boast straightforward designs and polarization control, thanks to the independent manipulation of individual meta-atoms. However, our attention shifts to narrowband meta-lenses with nonlocal responses, tailored for specific applications. Herein lies the challenge: striking a delicate balance between harnessing individual meta-atom responses for local phase control and navigating the complexities of nonlocal resonance excitation arising from interactions among neighboring meta-atoms. Presently, no established strategies exist to comprehensively address the myriad facets - Q-factor, efficiency, manipulation dimension, and footprint - simultaneously within the context of nonlocal wavefront shaping. Our work endeavors to tackle this conundrum with a novel approach, exemplified by a meticulously crafted metasurface. Let us delve deeper into the pioneering aspects and advancements of our methodology on the pioneering design of IRU (also refer to Supplementary Fig. 8).

(a) Amplified Q-factor: By disrupting the structure's in-plane symmetry, we usher in symmetry-protected q-BICs, lending a significant boost to the Q-factor. This augmentation is achieved by fine-tuning the asymmetric parameter, affording flexibility in manipulation.

(b) Stalwart rotation angle robustness: The rotation angle assumes paramount

importance in generating the geometric phase essential for meta-lens functionality. Ensuring the steadfastness of resonance characteristics - Q-factor, efficiency, phase, and resonant wavelength - across varying rotation angles is imperative for seamless phase modulation and resonance excitation.

(c) Enhanced transmission efficiency: Beyond the q-BIC mode, our innovation encompasses the incorporation of a Mie-type MD resonance. This synergistic coupling with q-BIC engenders the generalized Kerker effect, fostering unidirectional transmission and surpassing the theoretical limit of 25% for transmissive circular polarization conversion.

Concurrently attaining these three pivotal points presents a formidable challenge within the realm of conventional meta-units characterized by a single resonant mode and lackluster structure. Moreover, the optimization of IRUs warrants a delicate equilibrium, wherein geometric shape optimization necessitates a harmonious interplay between symmetry and asymmetry to cater to rotation robustness and the excitation of leaky q-BIC excitation. Selecting a lattice and unit with improved symmetry while maintaining asymmetry, such as a hexagonal lattice, optimizes this balance. The crescent shape, derived from a circle, is chosen for its superior symmetry, and the optimization process is detailed in Supplementary Fig. 9. Similarly, the optimization of array periods demands a judicious equilibrium, balancing weak coupling for rotation robustness with strong coupling between IRUs to realize a high Q-factor (refer to Supplementary Fig. 10)."

Supplementary Fig. 8: Design flow for an IRU with high Q-factor, rotation robustness, and high efficiency.

2. The manuscript does not provide a clear and comprehensive comparison with other similar works. For example, what is the advantage of this work over [Lin L., et al, Universal narrowband wavefront shaping with high quality factor meta-reflect-arrays. *Nano Lett.* 23, 1355-1362 (2023)] ? The design in that paper provides higher Q-factor and comparable efficiency when compared with this work.

Response: We thank the Reviewer for the constructive comment. For the work provided by the Reviewer [*Nano Lett.* 23, 1355-1362 (2023), i.e. Ref. [34] in updated main text], it proposed a novel phase gradient metasurface made entirely from individually addressable high-Q-factor meta-atoms. Although the Q-factor of ~ 380 is higher than our work and the efficiency of 59% is comparable, it should be noted that our approach offers several advantages over the referenced work. Furthermore, the underlying physics behind the metasurfaces differs significantly:

(a) Advantages

- i. With high Q-factor and efficiency, our work realizes the two-dimensional wavefront shaping, while the reported work only demonstrates the one-dimensional control. The optical performance will be significantly affected by the addition of wavefront shaping dimension, and the range of versatile applications will be constrained by the limited manipulation dimension.
- ii. Our work does not require extremely precise fabrication. As shown in Fig. 2b, the q-BIC mode and Kerker effect exhibit relative stability even with slight wavelength deviations when changing the geometric parameters. In contrast, the reported work, which manipulates phase shifts through structural perturbations, necessitates precise control over geometric parameters for wavefront engineering. This demands a robust fabrication process with precision up to a few nanometers, posing challenges for practical implementation.
- iii. For practical applications, transmissive optical components are more commonly used. In Ref. [34], a metallic back reflector is involved, limiting the metasurface to operate only in a reflection scheme, which significantly constrains its range of applications. In contrast, our metasurface operates in a transmission scheme with the capability of 2D wavefront control, making it more promising for real-world applications.

(b) Differences

The physical mechanisms of our work and this reported work are inherently distinct. In our approach, we leverage q-BIC and the Kerker effect to achieve both a high Q-factor and a high efficiency. The rotation robustness for the geometric phase is also optimized. In contrast, the reported work focuses

on manipulating the geometric parameters of meta-atoms to control the resonance phase of guide-mode resonances. Furthermore, the incident polarization states in these two works differ, with our work utilizing circular polarization and the reported work employing linear polarization. Despite the differences, both approaches offer a platform for efficient high-Q wavefront shaping.

Considering the combined aspects of Q-factor, efficiency, manipulation dimension, footprint, and fabrication requirements, our work demonstrates strong performance compared with other studies.

We have revised the relevant description to make the comparison clearer and more comprehensive on Pages 11 and 12 in **Main Article**:

“More comparisons with other similar works are given in Supplementary Table 1. In terms of Q-factor, efficiency, manipulation dimension, footprint, and fabrication requirements, our work stands out with superior performance when compared to previous studies (refer to Supplementary Note 14 for more details and discussions).”

and added a detailed comparison on Pages 19 and 20 in **Supplementary Note 14**:

“We have compared our work with Ref. [5] in the main text because both two works are single-layer nonlocal meta-lenses for two-dimensional wavefront shaping. To provide a more comprehensive comparison, a comparison with another similar work of Ref. [7] is also given as follows. It proposed a novel phase gradient metasurface made entirely from individually addressable high-Q-factor meta-atoms. Although the Q-factor of ~380 reported in Ref. [7] is higher than our work and the efficiency of 59% is comparable, it is important to highlight the advantages and distinctions of our work as follows:

(a) Advantages

- i. Our work achieves two-dimensional wavefront shaping with high Q-factor and efficiency, surpassing the previous work's demonstration of only one-dimensional control⁷. The addition of wavefront shaping dimension significantly impacts optical performance, while the limited manipulation dimension constrains versatile applications.
- ii. Unlike previous work, our fabrication process doesn't demand extreme precision. In Fig. 2b in the main article, the q-BIC mode and Kerker effect exhibit stability even with slight wavelength deviations in changing geometric parameters. In contrast, previous work requires precise control over geometric parameters for wavefront engineering⁵, posing challenges for practical implementation due to the need for a fabrication process with precision up to a few nanometers.
- iii. Transmissive optical components are more commonly used in practical applications. The involvement of a metallic back reflector in Ref. [7] confines the metasurface to operate only in a reflection scheme, limiting

its range of applications. In contrast, our metasurface operates in a transmission scheme with 2D wavefront control capability, making it more promising for real-world applications.

(b) Differences

The physical mechanisms between our work and previous work are inherently distinct. In our approach, we leverage q-BIC and the Kerker effect to achieve both high Q-factor and efficiency, along with optimized rotation robustness for the geometric phase. In contrast, the reported work manipulates the geometric parameters of meta-atoms to control the resonance phase of guide-mode resonances. Additionally, the incident polarization states differ, with our work utilizing circular polarization and the reported work employing linear polarization. Despite these differences, both approaches offer a platform for efficient high-Q wavefront shaping.”

3. The paper claims that the lens provides a sub-diffraction-limited full-width half-maximums (FWHMs). But it seems that this claim is only based on the results shown in the x-y plane (Fig. 4b). The data from the x-z plane (Fig. 4c) suggests that the FWHM is much larger in that direction. To clarify this point, I recommend that the authors plot the data from Fig. 4c in a 1D graph as well, as they did for the x-y plane.

Response: We thank the Reviewer’s constructive suggestion. In fact, the full-width half-maximums (FWHMs) in the xy plane signify sub-diffraction-limited performance. Conversely, in the xz plane, this measure is referred to as the depth of focus (DOF) rather than FWHM. The experimental, simulated, and theoretical DOFs are 61.2, 70.3, and 62.8 μm , respectively, as shown in Fig. R1. The deviation between these three results is due to imperfect amplitude, phase, and wavelength in practical operations resulting from resonance excitations and fabrication imperfection.

Fig. R1: Revised Fig. 4d with intensity distributions in the z direction.

We have added the intensity distribution in the z direction in Fig. 4d. Some discussions have been added on Page 13 in **Main Article**:

“Their full-width half-maximums (FWHMs) are 4.8, 4.1, and 3.9 μm , respectively, implying a sub-diffraction-limited performance ($\lambda/2\text{NA} = 3.9 \mu\text{m}$) on the focal plane. Corresponding depths of focus along the z direction are 61.2, 70.3, and 62.8 μm , respectively, as shown in Fig. 4d. The deviation between these three results is due to imperfect amplitude, phase, and wavelength in practical operations resulting from resonance excitations and fabrication imperfection.”

4. Comparing the experimental results reported in Fig. 4 of the manuscript to the simulation results reported in Fig. 10 of the supplementary, there is a considerable shift in the operating wavelength. Considering the fact that the lens is designed to be very high-Q and therefore narrowband, this shift is important and need to be addressed and explained in the manuscript.

Response: We thank the Reviewer’s comment on this point. The experimental resonant wavelength experiences a redshift of ~ 20 nm compared to the simulated wavelength, which is primarily attributed to the fabrication imperfection. As shown in Fig. 2b, the excitation of q-BIC mode remains relatively stable even with slight deviations in feature size, resulting only in a shift in resonant wavelength and a minor change in Q-factor. Additionally, the Kerker effect appears more resilient to the fabrication imperfections since the Mie-type MD resonance associated with it has a low Q-factor, leading only to a wavelength shift and a slight reduction in efficiency. Therefore, while there is a deviation in resonant wavelength, the phenomenon of wavelength-selective property remains largely unaffected, as evident in Fig. 4e and Supplementary Fig. 14.

To address the reviewer’s concern, we have added the following discussions and descriptions on Page 13 in **Main Article**:

“The only noticeable effect is a wavelength deviation of approximately 20 nm, while the overall wavelength-selective property remains largely unaffected.”

and on Page 18 in **Supplementary Note 13**:

“The experimental resonant wavelength undergoes a red shift of ~ 20 nm compared with the simulated wavelength, which is primarily due to the fabrication imperfection. Figure 2b demonstrates that the excitation of q-BIC mode remains relatively stable, even with slight deviations in feature size, resulting in only a slight shift of resonant wavelength and a minor change of Q-factor. Another crucial observation is that the Kerker effect appears more resilient to fabrication imperfections as the Mie-type MD resonance exhibits a

low Q-factor, leading only to a wavelength shift and a slight reduction in efficiency. Consequently, only a wavelength deviation of about 20 nm occurs, while the overall phenomenon of wavelength-selective property remains minimally affected, as evident in Fig. 4e and Supplementary Fig. 14.”

For Reviewer 3:**General Comment:**

The authors have done a solid job. However, some results and their interpretation raise questions. I do not recommend the manuscript for publication in the current form.

I encourage the authors to address the following questions:

Response:

We thank the Reviewer for his/her positive comments and constructive suggestions on our manuscript. We addressed the comments point-to-point as follows.

Specific Comments:

1. Could the Authors, briefly explained the difference local and non-local metasurfaces and also explain in simple words why the theoretical limit of polarization conversion is 25% for nonlocal metasurfaces and there is no such a limit for local metasurfaces? Please also defined the efficiency of polarization conversion in the text. The cited paper Ref. 35 does not explain the limit of 25% but it addresses authors the readers to other papers of the same authors.

Response:

We thank the Reviewer's comment. From the generalized concept, the main difference between local and nonlocal metasurfaces lies in whether the interactions between meta-atoms can be ignored or not. For local metasurfaces, each meta-atom is optically isolated, implying that the interaction with adjacent unit elements is negligible. For nonlocal metasurfaces, optical responses are influenced by both the individual responses of each meta-atom and their significant interactions with others. The collective behavior of several or numerous meta-atoms contributes to generating the desired functionality. While manipulating the nonlocal effect can be challenging, it provides an additional degree of freedom for customizing the electromagnetic responses of metasurfaces.

According to the specific definition in Fig. 1, another difference between local and nonlocal metasurfaces lies in the height of meta-atoms. When illuminated with circularly polarized light, local dielectric meta-lenses typically require meta-atoms with a considerable structural height (often close to the operating wavelength) to achieve high polarization conversion efficiency. On the other hand, nonlocal meta-lenses are typically designed to be optically thin (usually less than half of the wavelength), with minimal out-of-plane symmetry to induce nonlocal effects and strong resonance excitations.

Regarding the theoretical limit of polarization conversion efficiency, it stands at

25% for ultrathin metasurfaces. In fact, the transmission polarization conversion amplitude t_{LR} can be derived as

$$|t_{LR}|^2 = \text{Re}[t_{LL}] - |t_{LL}|^2 \quad (\text{R1})$$

where t_{ij} denoted the i -polarized transmission coefficient of the metasurface when it is illuminated with j -polarized light. The subscript L and R represent left-handed polarization and right-handed polarization, respectively. Referring to Eq. (R1), the maximum cross-polarization conversion efficiency $T_{LR} = |t_{LR}|^2$ reaches 25% when $t_{LL} = 50\%$. More theoretical demonstrations can be found in Refs. [39,43-45] in the main text. Figure 1 indicates the general cases of three types of meta-lenses: dielectric local meta-lens with wavelength-level height, nonlocal meta-lens with ultrathin design, and nonlocal Huygens' meta-lenses with optically thin thickness. Local dielectric meta-lenses typically need high-aspect-ratio meta-atoms to achieve a high polarization conversion efficiency. These systems are not optically thin and typically exhibit a broken out-of-plane symmetry, thereby avoiding the 25% efficiency limitation. On the other hand, nonlocal meta-lenses face challenges including nonlocal effects and strong resonance excitations in meta-atoms with a high aspect ratio. Hence, they are usually designed to be thin, almost without out-of-plane symmetry, with their polarization conversion efficiency limited to $\sim 25\%$.

To address the reviewer's concerns, we have revised the descriptions of the polarization conversion limitation on Pages 5 and 6 in the **Main Article**:

"When illuminated with broadband circularly polarized light, local dielectric meta-lens can focus the entire spectrum without wavelength selectivity. This capability is accompanied by a high transmission polarization conversion efficiency T_{LR} attributed to the high aspect ratio of its meta-atoms, which breaks the out-of-plane symmetry¹⁰. Here, T_{LR} is defined as the ratio between the powers of polarization-converted light (right-circularly polarized, RCP) to the incidence (left-circularly polarized, LCP). By harnessing the nonlocal effect arising from the interactions between meta-atoms, we achieve effective focusing solely on narrowband light, leaving light of non-resonant wavelengths unmodulated. Since introducing the nonlocal effect in meta-atoms with a high aspect ratio presents challenges, nonlocal meta-lenses are typically designed to be thin, lacking out-of-plane symmetry. Consequently, they function as a four-port system, with the transmissive circularly polarized conversion efficiency capped at $\sim 25\%$ ^{39,43-45} (refer to the middle image in Fig. 1). Further elaboration on the theoretical limit is provided in Supplementary Note 7."

with additional references

43. Monticone F., Estakhri N. M., Alu A. Full control of nanoscale optical transmission with a composite metascreen. *Phys. Rev. Lett.* **110**, 203903 (2013).

44. Ding X., Monticone F., Zhang K., Zhang L., Gao D., Burokur S. N., De Lustrac A., Wu Q., Qiu C. W., Alu A. Ultrathin Pancharatnam-Berry metasurface

with maximal cross-polarization efficiency. *Adv. Mater.* **27**, 1195-1200 (2015).
45. Hassanfiroozi A., Huang P. S., Huang S. H., Lin K. I., Lin Y. T., Chien C. F., Shi Y., Lee W. J., Wu P. C. A Toroidal-Fano-resonant metasurface with optimal cross-polarization efficiency and switchable nonlinearity in the near-infrared. *Adv. Opt. Mater.* **9**, 2101007 (2021).”

The caption of Fig. 1 is also revised to clarify the difference:

“For simplicity, the local meta-lens discussed in this work comprises dielectric meta-atoms with heights at the scale of wavelengths, while the nonlocal meta-lenses are optically thin.”

The following discussions regarding the theoretical limit of polarization conversion efficiency are added on Page 8 in **Supplementary Note 7**:

“As discussed in the main context, the metasurface can be conceptualized as a four-port system. In such cases, the transmission coefficient of cross-polarization conversion t_{LR} can be derived as^{1, 2}:

$$|t_{LR}|^2 = \text{Re}[t_{LL}] - |t_{LL}|^2 \quad (\text{S1})$$

Here, t_{ij} represents the transmission coefficient for i -polarized light when the metasurface is illuminated with j -polarized light. The subscripts L and R signify left-handed polarization and right-handed polarization, respectively. Referring to Eq. (S1), the maximum cross-polarization conversion efficiency $T_{LR} = |t_{LR}|^2$ reaches 25% when $t_{LL} = 50\%$.”

2. As I know, the overcoming the limit of the polarization conversion of 25% requires the breaking out-of-plane symmetry. Due to the substrate, the out-of-plane symmetry is already broken and, thus, this limit is not applicable to the structures in Fig. 2. Therefore, strictly speaking, Figure 2 is not correct.

Response: We thank the Reviewer for this comment. We agree that breaking the out-of-plane symmetry can indeed surpass the efficiency limit of 25%. In fact, two scenarios that fulfill this condition have been reported and investigated. The first, as pointed out by the reviewer, involves breaking out-of-plane symmetry due to the refractive index disparity between the meta-atoms and the substrate. To surpass the 25% conversion efficiency threshold using this method, guide-mode resonance within the substrate must be engaged [*Light: Sci. Appl.* **9**, 23 (2020)]. However, individually controlling the phase of each meta-atom presents a considerable challenge, as resonance occurs both at the level of individual meta-atoms and within guided waves, with the latter constraining individual phase modulation. In the second scenario, the height of the meta-atom in local dielectric meta-lenses typically needs to reach the wavelength level. This specific setup facilitates the generation of waveguide-like modes inside the high-index meta-atoms, thereby enabling precise phase

control using individual units and overcoming limitations on polarization conversion efficiency (refer to the left panel in Fig. 1).

In our work, we neither introduce guide-mode resonance inside the substrate nor utilize meta-atoms with wavelength-level height. Therefore, our proposed approach does not rely on breaking out-of-plane symmetry to surpass the 25% polarization conversion efficiency threshold. The dominance of vertical magnetic dipole (MD) moments in the quasi-bound state in the continuum (q-BIC) mode, primarily contributed by the in-plane current, also supports our assertion: the out-of-plane symmetry break in our design is minimal. To further demonstrate that our proposed approach for achieving >25% polarization conversion efficiency in an ultrathin metasurface is attributable to the involvement of both the MD and q-BIC, we provide the polarization conversion efficiencies T_{LR} for various offsets L that controls the spectral distance between MD and q-BIC. As shown in Fig. 2b in the main article and Supplementary Fig. 7, when the offset L falls between 270 nm and 400 nm, the resonant wavelengths of the Mie-type MD resonance and the q-BIC mode approach. Two resonances couple with each other, resulting in the Kerker effect. At this point, the transmission polarization conversion efficiency surpasses 25%, reaching a maximum of 85.4% with $L = 310$ nm. However, when L exceeds 400 nm or is less than 270 nm, the two resonant wavelengths are significantly apart, and the single mode of q-BIC can only achieve an efficiency limited to around 25%. If the designed unit possesses an intrinsically large out-of-plane asymmetry, the efficiencies for all offsets should not be limited by 25%, and the rapid increase around $L = 310$ nm would not occur. These observations confirm the intrinsic weak out-of-plane asymmetry of the designed unit and underscore the importance of the Kerker effect.

To clarify this point, we have added the following discussions on Page 8 in the **Main Article** along with an additional figure of Supplementary Fig. 7:

“To better elucidate the achievement of polarization efficiency surpassing 25%, it is crucial to note that it is attributed to the synergistic engagement of both the Mie-type MD resonance and the q-BIC mode, rather than the disruption of out-of-plane symmetry. We explore this by varying the structural parameter of L to modulate the resonant wavelengths of MD resonance and the q-BIC mode. As shown in Supplementary Fig. 7, we observe that the conversion efficiency remains capped at ~25% when the resonances of MD and q-BIC are spectrally distant, indicating the inherent weakness of out-of-plane asymmetry. More discussions about the out-of-plane symmetry can be found in Supplementary Note 8.”

We also added the following discussions to the Supplementary.

Supplementary Note 8:

“In the realm of dielectric metasurfaces, disrupting out-of-plane symmetry has

been identified as a means to surpass the 25% efficiency threshold. In fact, two scenarios meeting this criterion have been explored. The first scenario involves breaking out-of-plane symmetry, which arises from the difference in refractive index between the meta-atoms and the substrate. To exceed the 25% conversion efficiency threshold using this approach, activation of resonances in the substrate like guide-mode resonance is necessary³. However, achieving precise control over the phase of each meta-atom poses a significant challenge, as resonance occurs both at the individual meta-atom level and within guided waves, with the latter restricting individual phase modulation. Alternatively, in the second scenario, the height of meta-atoms in local dielectric meta-lenses typically needs to match the wavelength level⁴. This configuration facilitates the generation of waveguide-like modes inside the high-index meta-atoms, enabling precise phase control using individual units and overcoming limitations on polarization conversion efficiency (see the left panel in Fig. 1).

In this work, we abstain from introducing guide-mode resonance within the substrate or utilizing meta-atoms with wavelength-level height. Consequently, our proposed approach does not hinge on breaking out-of-plane symmetry to surpass the 25% polarization conversion efficiency threshold. Furthermore, the predominance of vertical magnetic dipole (MD) moments in the quasi-bound state in the continuum (q-BIC) mode, primarily contributed by the in-plane current, lends support to our claim: the disruption of out-of-plane symmetry in our design is minimal. To further illustrate that our proposed approach for achieving >25% polarization conversion efficiency in an optically thin metasurface is attributable to the involvement of both the MD and q-BIC, we present the polarization conversion efficiencies T_{LR} for various offsets L that control the spectral separation between MD and q-BIC. As can be seen in Fig. 2b in the main article and Supplementary Fig. 7, when the offset L ranges between 270 nm and 400 nm, the Mie-type MD resonance couples with the q-BIC mode, resulting in the generalized Kerker effect. Consequently, the transmission polarization conversion efficiency exceeds 25%, reaching a maximum of 85.4% with $L = 310$ nm. However, when L exceeds 400 nm or falls below 270 nm, the two resonant wavelengths are significantly distant, and the single mode of q-BIC can only achieve an efficiency limited to approximately 25%. If the designed unit possesses an intrinsically large out-of-plane asymmetry, the efficiencies for all offsets should not be limited to 25%, and the rapid increase around $L = 310$ nm would not occur. These observations confirm the intrinsic weak out-of-plane asymmetry of the designed unit and underscore the importance of the Kerker effect.”

Supplementary Fig. 7: Transmission T_{LR} of q-BICs at resonant wavelengths as a function of offset L .

3. The inset in Fig. 1 illustrating the fabricated sample is unclear. It is quite difficult to understand the difference between the initial design and the fabricated sample.

Response: We thank the Reviewer’s comment on this point. To make the titled scanning electron microscope (SEM) image clearer, we have selected a clearer area of the sample for SEM image display. We have also improved the image resolution and zoomed in the image, as shown in Fig. R2. More SEM images have been shown in Fig. 3a-c.

Fig. R2: Revised Fig. 1 with a clearer SEM image of the fabricated sample.

4. The design of metasurface is unclear from the provided figures. If I understood correctly, the structure is periodic, and all the meta-atoms are identical. The only difference between the unit cells of the metalens is the

rotation angle of the crescent shape meta-atoms. It would be illustrative if the authors encode the orientation of the crescent shape meta-atoms with colors and plot the design of the metalens (top view).

Response: We would like to express our gratitude to the Reviewer for assisting in enhancing the clarity of our paper. The layout of meta-lens units with encoded orientation angle is shown in Fig. R3, which has been added as Fig. 4a to align with the optical microscopic image of the fabrication sample. Relevant descriptions have been added to Page 13 in the **Main Article** to provide further context:

“Figure 4a represents the layout of meta-lens units with encoded rotation angles, as per Eq. (1). Corresponding to Fig. 4a, Fig. 4b shows the optical microscopic image of the meta-lens, featuring a diameter of 90 μm .”

Fig. R3: The arrangement of meta-lens units with encoded rotation angles.

5. Why do the Authors need q-BIC and high Q-factor? How does the high-Q factor affect the performance of metalens? High-Q factor shows the near-field enhancement but does not affect the far-field field enhancement. Why do the Authors need near-field enhancement due to quasi-BIC and high-Q factor?

Response: We thank the Reviewer's comment. High-Q-factor meta-lens and wavefront shaping are necessary because:

(a) Practical applications. High-Q-factor (narrowband) meta-lenses combine the properties of meta-lens and narrowband filters, catering to specific applications beyond conventional or broadband meta-lenses. These applications include nonlinear generation, bio-medical imaging, augmented reality/virtual reality display (AR/VR), *etc.* For example, nonlinear meta-lenses necessitate both high-Q-factor resonance and phase modulation capability. Biomedical imaging relies on specific wavelengths to match the spectral responses of biomolecules. AR/VR applications demand multicolor imaging at specific wavelengths, such as red, green, and blue, as demonstrated in Supplementary Fig. 15.

- (b) Additional degrees of freedom of spectra. In addition to near-field enhancement, q-BIC, and high Q-factor characteristics introduce additional spectral control with spatial phase modulation, which remains nearly stable. This feature offers more degrees of freedom for efficient and multifunctional meta-devices.
- (c) Challenge in the field of nonlocal meta-devices. Nonlocal meta-lenses with high Q-factor face the challenge of striking the right balance between local phase control, driven by the independent response of each meta-atom, and nonlocal resonance, resulting from the interactions between neighboring meta-atoms. While numerous studies have aimed to achieve high-Q-factor wavefront shaping, to the best of our knowledge, no established strategies effectively address all these aspects - Q-factor, efficiency, manipulation dimension, and footprint - simultaneously within the realm of nonlocal wavefront shaping.

Building on these points, we believe that achieving a high-Q-factor meta-lens utilizing nonlocal effects could indeed lead to significant advancements in nanophotonics and advanced optical systems. While various potential approaches exist for realizing high-Q responses, such as Fano resonance [*Nat. Photonics* 11, 543-554 (2017)], non-radiating anapole mode [*ACS Nano* 12, 1920-1927 (2018)], micro-cavity-integrated nanophotonics [*Optica* 10, 269-278, (2023)], and coherent lattice resonance [*Nanophotonics* 11, 2701-2709, (2022)], none of these methods simultaneously achieve both high-Q response and phase shift modulation, which is the primary focus of our work.

To provide further insight into the significance of q-BIC and high Q-factor, additional descriptions have been included and revised on Pages 3 and 4 in **Main Article:**

“In fact, with the inclusion of additional spectral control and stable spatial phase modulation, there is significant interest in narrowband-operated wavefront shaping driven by nonlocal responses. Such advancements find applications in various fields including optical nonlinearity, biomedical imaging, thermal radiation, and augmented reality/virtual reality display²³⁻²⁸. While many studies have successfully demonstrated high Q-factor responses associated with high field enhancement using metasurfaces, leveraging phenomena such as Fano resonance²⁹, non-radiating anapole mode³⁰, micro-cavity-integrated nanophotonics³¹, and coherent lattice resonance³², none of these methods have been able to simultaneously achieve both high-Q response and phase shift modulation.”

and on Page 21 in **Supplementary Note 15:**

“The proposed nonlocal Huygens’ meta-lens can be extended to the visible band, providing potential applications in multispectral color routers and augmented reality/virtual reality displays, as shown in Supplementary Fig. 15.

Furthermore, the nonlocal Huygens' meta-lens holds promise in nonlinear generation and biomedical imaging, surpassing the capability of conventional or broadband meta-lenses. For example, nonlinear meta-lenses need both high-Q-factor resonance and phase modulation capability, while biomedical imaging demands specific wavelengths to match the spectral responses of biomolecules.”

6. Vertical and horizontal magnetic dipoles cannot result in the enhanced scattering to forward direction as the vertical magnetic dipole does not radiate to the vertical direction. Therefore, it cannot affect the transmission at normal incidence.

Response: We thank the Reviewer for this comment. Introducing the vertical MD (MD_z) serves to elucidate the resonance characteristics of the q-BIC mode and distinguish it from the low-Q horizontal MD (MD_{xy}) resonance. The vertical MD (MD_z) resonance (q-BIC) is originally a dark mode for normal incidence. Through the breaking of symmetry in the parameter space, the q-BIC mode is effectively induced by asymmetric horizontal dipole moments. While the vertical MD itself does not generate vertical radiation, other horizontal multipole moments induced by the q-BIC mode can influence normal transmission, such as electric dipole, toroidal dipole, *etc.*, as evidenced by peaks in the scattering spectra of horizontal multipoles (see Supplementary Fig. 11). These multipoles interfere with the horizontal MD induced by the Mie-type MD resonance, creating the generalized Kerker effect that disrupts the radiation symmetry and enhances the transmission efficiency. This phenomenon is also reflected in the radiation pattern represented in Fig. 2e. Additionally, considering the influence of toroidal dipole response, the scattering spectra in Fig. 2d have been modified in accordance with Fig. R4, which does not alter the underlying physical mechanism.

To clarify this point, the following discussions are added on Page 8 in **Main Article:**

“This inheritance of characteristics presents the potential for generating interference between dominant MD and other multipole modes radiating to the forward direction, ultimately leading to the emergence of the generalized Kerker effect⁴⁶.”

and on Pages 9 and 10 in **Main Article:**

“Although out-of-plane MD cannot generate vertical radiation, other multipole moments induced by the q-BIC mode can influence the normal transmission, such as electric dipole, toroidal dipole, *etc.*, as evidenced by peaks in the scattering spectra in Fig. 2d. The scattering spectra of horizontal multipoles are also given in Supplementary Fig. 11. These multipoles interfere with the in-

plane MD induced by the low-Q MD resonance, creating the generalized Kerker effect that disrupts the radiation symmetry and enhances the transmission efficiency.”

Fig. R4: Revised Fig. 2d with multipole decomposition of an IRU with $L = 310$ nm.

Additional descriptions are added on Page 15 in **Supplementary Note 10:**

“The vertical MD (MD_z) resonance (q-BIC) is originally a dark mode for normal incidence. By breaking the symmetry in the parameter space, the q-BIC mode is effectively induced by asymmetric horizontal dipole moments. The vertical MD cannot generate vertical radiation, but other horizontal multipole moments induced by the q-BIC mode can influence the normal transmission, such as electric dipole, toroidal dipole, etc., which can be found by the peak in the scattering spectra of horizontal multipoles in Supplementary Fig. 11. These multipoles interfere with the horizontal MD induced by the Mie-type MD resonance, generating the generalized Kerker effect to break the radiation symmetry and improve the transmission efficiency.”

Supplementary Fig. 11: Normalized scattering power of horizontal multipoles of an IRU with $L = 310$ nm.

7. Figure 2b shows the transmission map. The maximum transmission is about 85%, but this value is almost indistinguishable in the figure.

Response: We thank the Reviewer for the helpful suggestion. The transmission spectra with $L = 200, 250, 310, 350,$ and 400 nm are given in Supplementary Fig. 5. The maximum transmission $T_{LR} = 85.4\%$ can be obtained for $L = 310$ nm at the resonant wavelength of ~ 1550 nm.

We have added the following sentence on Page 8 in the **Main Article** along with an additional figure of Supplementary Fig. 5:

“Detailed transmission spectra are given in Supplementary Fig. 5.”

Supplementary Fig. 5: Transmission spectra with $L = 200, 250, 310, 350,$ and 400 nm.

In addition, we have reduced the line widths of dashed lines and circles to make the spectra clearer in Fig. 2b. As can be seen, the maximum transmission emerges at the position of the Kerker effect for $L = 310$ nm, as indicated by one of the white circles (see Fig. R5).

Fig. R5: Revised Fig. 2b with reduced line widths of dashed lines and circles.

To indicate the point of maximum transmission clearer, the following sentence has been added in the caption of Fig. 2b on Page 9 in **Main Article**:

“The white circle of the Kerker effect is extracted from the wavelength with the highest efficiency for $L = 310$ nm.”

8. Am I right that Figure 2 is plotted for periodic structure but not for the metalens? This is unclear from the text of the paper.

Response: We thank the Reviewer for this comment. Indeed, Fig. 2 presents the design outcomes of the periodic unit rather than the meta-lens. Due to the careful design of the periodic unit (further elaborated in Response to Comment 9 from Reviewer 3), the meta-lens exhibits a similar optical response with the unit, which can be observed through the comparison between Fig. 3d-f and Supplementary Fig. 4.

More descriptions have been added and revised to make it clearer on Page 7 in **Main Article**:

“The optical responses shown in Fig. 2 are solely based on simulations of periodic IRUs.”

and Page 11 in **Main Article**:

“The simulated spectra of the corresponding periodic IRUs are presented in Supplementary Fig. 4.”

More comparisons and discussions between the optical responses of periodic integrated-resonant units (IRUs) and meta-lens have been added on Page 5 in **Supplementary Note 4**:

“Considering the arrangement of meta-atoms with different orientations in corresponding positions within the meta-lens, it is crucial to evaluate the rotation robustness of these meta-atoms, as demonstrated in Fig. 2g. This property indicates that the resonance attributes of meta-atoms, including resonant wavelength, efficiency, and phase shift, remain constant across varying rotation angles. Consequently, the nonlocal effect of interactions between meta-atoms is minimally affected, leading to a meta-lens akin to the designed periodic array. Nevertheless, despite the careful design of units, the meta-lens still experiences performance reduction compared to the inherent nature of the nonlocal effect. This difference is evident in the comparison between Fig. 3d-f and Supplementary Fig. 4. The simulated Q-factors and efficiencies of three meta-lens (periodic array of units) are 120 (205), 110 (144), and 71 (89), with corresponding efficiencies of 70% (85%), 72% (83%), 69% (80%), respectively. Further optimization of performance may be achieved by leveraging artificial intelligence technology to refine the interactions between meta-atoms with different orientations.”

9. What is the physical mechanism responsible for the polarization conversion? How was circular polarization transmission $T_{\{LR\}}$ optimized to achieve the maximal value? Blind numerical optimization?

Response: We thank the Reviewer for this comment. The increase in polarization conversion is primarily attributed to the generalized Kerker effect, characterized by the effective coupling and interference between the q-BIC and Mie-type MD resonances. This interaction is essentially governed by the geometric parameter of offset, denoted as L . Furthermore, the introduction of offset disrupts structural symmetry, thereby introducing an additional degree of freedom to induce orthogonal dipole moments and high-order multipoles. This multifaceted coupling mechanism facilitates the realization of polarization conversion functionality. Following an understanding of the underlying physical mechanisms, further numerical optimizations are necessary to fine-tune the optical performance of the meta-atom.

To elucidate the physical mechanism behind the polarization conversion, the following discussions are included on Page 8 in **Main Article**:

“This phenomenon is primarily governed by the geometric parameter known as offset L . Furthermore, the introduction of offset disrupts structural symmetry, thereby introducing another level of freedom to generate orthogonal dipole moments and high-order multipoles. This complex coupling condition facilitates the achievement of polarization conversion functionality.”

10. Usually, the Q-factor of q-BIC decreases with the increase in the asymmetry parameter but Figure 2f shows the inverse behavior. What is the reason for such a behavior? Is this resonance indeed a q-BIC? What are the benefits from a metalens with high-Q factor?

Response: We thank the Reviewer for this comment. For offset parameter L greater than 250 nm, as the offset increases, the structural asymmetry diminishes, resulting in a higher Q-factor for the q-BIC. The sudden change observed when the offset L is 200 nm is attributed to the influence of Wood's anomaly mode [*Nat. Commun.* 4, 2381 (2013)], which occurs at a wavelength of ~ 1256 nm. The theoretical resonant wavelength can be calculated using the formula

$$\lambda_{\text{WA}} = n_s P,$$

where $n_s = 1.45$ is the refractive index of the substrate, and $P = 1000\sqrt{3}/2$ nm is the effective period of the hexagonal lattice. Since this phenomenon does not

impact our physical mechanism and target functionality, it is omitted for simplicity. We have revised the descriptions on Page 10 in **Main Article**:

“The abrupt shift observed at $L = 200$ nm is attributed to the influence of Wood’s anomaly mode⁵⁰, which occurs at a wavelength of ~ 1256 nm. However, as this phenomenon does not impact our target functionality, it is omitted from further discussion for simplicity.”

Regarding the benefits of high-Q-factor meta-lens, as discussed in Response to Comment 5 by the same reviewer, it is noted that these meta-lenses offer distinct potential advantages. In addition to near-field enhancements, they afford an extra degree of freedom to engineer the wavefront and optical spectra. Further elaboration on this aspect can be found in Response to Comment 5.

11. Can the authors plot directivity in Fig. 2D, i.e. intensity along each direction normalized by the intensity averaged over all directions?

Response: We thank the Reviewer for this comment. Figure 2d shows the multipole decomposition, illustrating the characteristics of two resonances. In Fig. 2e, the power directivities are presented for $L = 200, 310, 400,$ and 500 nm, demonstrating the evolutionary process of the Kerker effect. As suggested, the power has been normalized by the averaged power over all directions, as shown in Fig. R6. The directivities [ACS Nano 14, 15042-15055 (2020)] along the +z direction are 1.26, 4.37, 2.34, and 1.41, respectively. We have rectified the inaccuracy in the far-field radiation power calculation, which does not affect our physical mechanism or other results.

Fig. R6: Revised Fig. 2e with far-field radiation patterns in the xz plane for IRUs with $L = 200, 310, 400,$ and 500 nm. The power has been normalized by the averaged power over all directions.

We have revised Fig. 2e and added more discussions on Page 10 in **Main Article**:

“Figure 2e shows the radiation pattern of the IRU with various offsets $L,$

demonstrating the evolutionary process of the Kerker effect. The power has been normalized by the averaged power over all directions.”

and on Page 10 in **Main Article**:

“The directivities⁴⁸ along the +z direction for $L = 200, 310, 400,$ and 500 nm are 1.26, 4.37, 2.34, and 1.41, respectively.”

12. The nonlocal effect, which leads to the generation of Q-BIC, plays a crucial role in the proposed nonlocal Huygens' metalens. However, this effect is highly sensitive to the surrounding nanostructures. To illustrate the nonlocal effect in the proposed nonlocal Huygens' metalens, the authors analyze the multipole modes supported in the periodic nanostructure with varying radial unit numbers. Nevertheless, if these nanostructures are employed to create a metalens with a non-periodic array due to the geometric phase, how do the nanostructures with different orientations affect the generation of Q-BIC and the working wavelength?

Response: We thank the Reviewer for this comment. The nonlocal effects have been demonstrated in Supplementary Fig. 6. In Supplementary Figs. 6a and 6b, a comparison is presented between a periodic array and a single unit, illustrating the nonlocal effect of the meta-unit stemming from the q-BIC mode. Furthermore, Supplementary Fig. 6c delves into the analysis of the nonlocal effect based on the radial unit number, which is determined by the meta-atoms' varied structural orientations instead of a periodic array. As can be seen in Supplementary Fig. 6c, the Q-factor experiences a significant increase with the augmentation of meta-units along the radial direction, signifying the attainment of q-BIC. We have added the relevant description to make it clearer on Page 7 in **Supplementary Note 6**:

“The relationship between the Q-factor and the number of meta-units along the radial axis in the meta-lens is provided in Supplementary Fig. 6c. It is evident that the Q-factor rises with the increase in the number of meta-units along the radial direction, highlighting the presence of a nonlocal effect within the meta-lens, even when the building blocks possess different structural orientations.”

Considering the integration of meta-atoms with varying orientations in the meta-lens, it is crucial to assess the rotation robustness of these meta-atoms, as demonstrated in Fig. 2g. This property ensures that the resonance characteristics of meta-atoms, such as resonant wavelength, efficiency, and phase, remain stable across different rotation angles. Consequently, the nonlocal effect of the interaction between meta-atoms is minimally influenced, maintaining a response in the meta-lens akin to that of the designed periodic array. Further design details are given in Response to Comment 9. Nevertheless, despite the careful design of units, the meta-lens still exhibits performance reduction compared to the periodic array due to the inevitable

impact of meta-atom orientations and finite meta-lens size. This disparity can be observed through the comparison between Fig. 3d-f and Supplementary Fig. 4. The simulated Q-factor and efficiencies of three meta-lens (periodic array) are as follows: 120 (205), 110 (144), and 71 (89), with corresponding efficiencies of 70% (85%), 72% (83%), 69% (80%), respectively. Enhancing performance may be achieved by leveraging artificial intelligence technology to further optimize the interactions between meta-atoms with different orientations. We have added relevant descriptions on Page 5 in **Supplementary Note 4**:

“Considering the arrangement of meta-atoms with different orientations in corresponding positions within the meta-lens, it is crucial to evaluate the rotation robustness of these meta-atoms, as demonstrated in Fig. 2g. This property indicates that the resonance attributes of meta-atoms, including resonant wavelength, efficiency, and phase shift, remain constant across varying rotation angles. Consequently, the nonlocal effect of interactions between meta-atoms is minimally affected, leading to a meta-lens akin to the designed periodic array. Nevertheless, despite the careful design of units, the meta-lens still experiences performance reduction compared to the inherent nature of the nonlocal effect. This difference is evident in the comparison between Fig. 3d-f and Supplementary Fig. 4. The simulated Q-factors and efficiencies of three meta-lens (periodic array of units) are 120 (205), 110 (144), and 71 (89), with corresponding efficiencies of 70% (85%), 72% (83%), 69% (80%), respectively. Further optimization of performance may be achieved by leveraging artificial intelligence technology to refine the interactions between meta-atoms with different orientations.”

13. Is it possible to expand the applicability of the proposed nonlocal Huygens' metalens to work under ambient illumination? This is crucial considering the potential applications of the color router as shown in Supplementary Fig. 11: which typically operates in natural environments under ambient conditions.

Response: We thank the Reviewer for this comment. The updated Supplementary Fig. 15 demonstrated the potential applications of the nonlocal Huygens' meta-lens in multispectral color routers and AR/VR displays. The simulation utilizes collimated coherent light for illumination.

Under the ambient condition of uncollimated incoherent light, as shown in the newly added Supplementary Fig. 13, two resonances can be observed in the spectra. However, the performance metrics of Q-factor and efficiency for the nonlocal meta-lens are diminished. Addressing the dependency on collimation and coherence necessitates further design and optimization of both local and nonlocal responses to enable the effective operation of the meta-lens in natural environments. For instance, it is essential to account for the coherence length

of ambient light and the diameter of the meta-lens. This ensures that resonance excitation and coupling between meta-atoms are not excessively influenced.

To discuss this point, we have added the following descriptions on Page 12 in the **Main Article**:

“Indeed, both the collimation and coherence of incident light can significantly impact optical performance. Supplementary Fig. 13 offers further discussions on the collimation and coherence of incident light, specifically in relation to the optical performance of the meta-lens.”

and on Page 17 in **Supplementary Note 12**:

“Supplementary Fig. 13 represents the measured spectra of three nonlocal Huygens’ meta-lenses illuminated under ambient conditions of uncollimated incoherent light (OCTW-UV-VIS-NIR from Oceanhood). Despite the presence of two resonances in the spectra, the Q-factor and efficiency performances of the nonlocal meta-lens are diminished. Addressing the dependence on collimation and coherence phenomena requires additional design and optimization of both local and nonlocal responses. This optimization is necessary to enhance the effectiveness of the meta-lens in operating within natural environments.”

REVIEWERS' COMMENTS

Reviewer #2 (Remarks to the Author):

The authors have addressed all of my concerns in the revised manuscript. Therefore, I recommend the revised manuscript for publication.

Reviewer #3 (Remarks to the Author):

The Authors have addressed all my questions and concerns. I recommend the manuscript for acceptance in its current form.